# The Fit After Baby randomized controlled trial: An mHealth postpartum lifestyle intervention for women with elevated cardiometabolic risk

**Jacinda M. Nicklas**[1]*, **Laura Pyle**[2,3], **Andrey Soares**[1], **Jennifer A. Leiferman**[4], **Sheana S. Bull**[4], **Suhong Tong**[3], **Ann E. Caldwell**[5,6], **Nanette Santoro**[7], **Linda A. Barbour**[5,6]

1 Department of Medicine, Division of General Internal Medicine, University of Colorado School of Medicine, Aurora, Colorado, United States of America, 2 Department of Pediatrics, University of Colorado School of Medicine, Aurora, Colorado, United States of America, 3 Department of Biostatistics and Informatics, University of Colorado School of Public Health, Aurora, Colorado, United States of America, 4 Department of Community and Behavioral Health, University of Colorado School of Public Health, Aurora, Colorado, United States of America, 5 Department of Medicine, Division of Endocrinology, Metabolism, and Diabetes, University of Colorado School of Medicine, Aurora, Colorado, United States of America, 6 Department of Obstetrics and Gynecology, Division of Maternal-Fetal Medicine, University of Colorado School of Medicine, Aurora, Colorado, United States of America, 7 Department of Obstetrics and Gynecology, Division of Reproductive Endocrinology and Infertility & Reproductive Sciences, University of Colorado School of Medicine, Aurora, Colorado, United States of America

* Jacinda.Nicklas@cuanschutz.edu

## Abstract

### Background

Postpartum women with overweight/obesity and a history of adverse pregnancy outcomes are at elevated risk for cardiometabolic disease. Postpartum weight loss and lifestyle changes can decrease these risks, yet traditional face-to-face interventions often fail. We adapted the Diabetes Prevention Program into a theory-based mobile health (mHealth) program called Fit After Baby (FAB) and tested FAB in a randomized controlled trial.

### Methods

The FAB program provided 12 weeks of daily evidence-based content, facilitated tracking of weight, diet, and activity, and included weekly coaching and gamification with points and rewards. We randomized women at 6 weeks postpartum 2:1 to FAB or to the publicly available Text4baby (T4B) app (active control). We measured weight and administered behavioral questionnaires at 6 weeks, and 6 and 12 months postpartum, and collected app user data.

### Results

81 eligible women participated (77% White, 2% Asian, 15% Black, with 23% Hispanic), mean baseline BMI 32±5 kg/m² and age 31±5 years. FAB participants logged into the app a median of 51/84 (IQR 25,71) days, wore activity trackers 66/84 (IQR 43,84) days, logged weight 17 times (IQR 11,24), and did coach check-ins 5.5/12 (IQR 4,9) weeks. The COVID-19 pandemic interrupted data collection for the primary 12-month endpoint, and impacted

**Data Availability Statement:** Because patient-level (e.g. "row-level" or "line-level") data is more readily

**Funding:** Author JMN was supported by three grants: - NIH BIRCWH K12 HD057022, National Institutes of Health, URL: https://orwh.od.nih.gov/career-development-education/building-interdisciplinary-research-careers-in-womens-health-bircwh NIH NHLBI 1K23HL133604, National Heart Lung and Blood Institute, https://www.nhlbi.nih.gov/grants-and-training/training-and-career-development/early-career - NIH/NCATS Colorado CTSA UL1 TR002541, National Institutes of Health, National Center for Advancing Translational Sciences, URL: https://ncats.nih.gov/ctsa The funders had no role in study design, data collection and analysis, decision to publish, or preparation of the manuscript.

**Competing interests:** The authors have declared that no competing interests exist.

diet, physical activity, and body weight for many participants. At 12 months postpartum women in the FAB group lost 2.8 kg [95% CI -4.2,-1.4] from baseline compared to a loss of 1.8 kg [95% CI -3.8,+0.3] in the T4B group (p = 0.42 for the difference between groups). In 60 women who reached 12 months postpartum before the onset of the COVID-19 pandemic, women randomized to FAB lost 4.3 kg [95% CI -6.0,-2.6] compared to loss in the control group of 1.3 kg [95% CI -3.7,+1.1] (p = 0.0451 for the difference between groups).

## Conclusions

There were no significant differences between groups for postpartum weight loss for the entire study population. Among those unaffected by the COVID pandemic, women randomized to the FAB program lost significantly more weight than those randomized to the T4B program. The mHealth FAB program demonstrated a substantial level of engagement. Given the scalability and potential public health impact of the FAB program, the efficacy for decreasing cardiometabolic risk by increasing postpartum weight loss should be tested in a larger trial.

## Introduction

Adverse pregnancy outcomes (APOs) provide an early warning of future cardiometabolic risk [1, 2], often before traditional risk factors for diabetes and cardiovascular disease (CVD) are detected [1]. Preeclampsia, preterm delivery, delivery of a small-for-gestational age (SGA) neonate, gestational hypertension, and gestational diabetes mellitus (GDM) are independently associated with a 50–300% increased risk for CVD in later life [3, 4]. Women with pregnancies complicated by GDM have a ~50% increased risk for developing type 2 diabetes mellitus (T2DM) within 10 years, develop atherosclerosis earlier [5], and have increased risk for hypertension [6] and CVD, as compared to women with non-GDM pregnancies [1, 2]. A history of preeclampsia also increases a woman's risk for T2DM [7, 8]. Nearly 30% of parous US women will have at least one of these predictive conditions during pregnancy [1]. Retention of weight gained during gestation is a major contributor to adult weight gain in women and contributes to their cardiometabolic risk. Even as little as 1 kg of postpartum weight retention is linked to further weight gain and the development of T2DM [9]. Excess body weight increases risk for CVD and T2DM in women at every age and in every ethnic group, by 40% for overweight and by as much as 300–400% for severe obesity [10].

Despite guidelines to achieve a healthy weight after delivery, engage in regular moderate to vigorous physical activity (MVPA), and eat a healthy diet [11–14], studies of women with a history of APOs show that they do not engage in risk reduction behavior more than women without a history of pregnancy complications [15, 16]. In fact, one study of at-risk women found that more gained than lost weight after their affected pregnancy, suggesting an urgent need for support for lifestyle modification [17, 18]. Pregnancy weight retained beyond 6–12 months postpartum is usually retained long-term and is a powerful independent risk factor for future obesity [19]. Given the significance of postpartum weight retention, the postpartum year is a critical window of opportunity to make lifestyle changes to decrease future risk of obesity and cardiometabolic disease [1, 20, 21], as well as APOs in subsequent pregnancies. In one study an increase of 1–2 BMI units between pregnancies was associated with a 20–40% increased risk of GDM and gestational hypertension, while a loss of 12 lbs between pregnancies decreased the risk for GDM by 75% [22, 23].

Postpartum women describe multiple barriers to face-to-face participation in risk reduction interventions, including time constraints, infant and breastfeeding demands, older childcare responsibilities, and reluctance to spend time away from family [24, 25]. However, postpartum women are heavy users of smartphones and show interest in health related apps [26], across race and ethnicity [27], which poses an opportunity for a customized intervention. Employing mobile technology for health promotion using mobile health (mHealth) is an innovative approach for high-risk women with multiple family/work demands [28]. Using an mHealth lifestyle intervention for this population leverages the widespread adoption of mobile devices among women of reproductive age and offers the potential of a scalable and cost-effective program that could extend health promotion into home and daily life. Scalable programs have potential for a substantial public health impact, even if effect sizes are modest [29].

Although some mHealth programs have been tested for postpartum weight loss [30], few have been developed specifically for women with recent APOs [31, 32]. Women with recent APOs may benefit from an intervention specifically addressing their pregnancy complications and increased risk as part of a comprehensive lifestyle program. We previously developed an mHealth intervention called Fit After Baby (FAB) [33]. We adapted the content from the Diabetes Prevention Program [34] to target postpartum women at elevated cardiometabolic risk due to a history of APOs, including women with a history of GDM, preeclampsia, gestational hypertension, pre-term birth, or delivery of an SGA baby. The intervention integrates multiple behavior change techniques known to be effective for behavior change among postpartum women [35], is designed with concepts from user-centered design and mobile technology in health promotion, and includes a gamification component with modest incentives [36–39]. FAB builds upon current evidence specific to the postpartum period, including: recommended weight loss after pregnancy [40], breastfeeding [41, 42], exercise [13], and the effects of diet and physical activity (PA) on breastfeeding [43]. We refined the FAB program through an iterative beta-testing process [33]. The primary objective of this study was to test the feasibility, acceptability, and preliminary efficacy of Fit After Baby in a randomized controlled trial.

## Methods

### Participants

We recruited women between 18 and 45 years of age with a postpartum BMI of 26 to 45 kg/m$^2$ ($\geq$24 for Asians based on their greater cardiometabolic susceptibility at a lower BMI [44]) who were between 4–12 weeks postpartum from a recent singleton delivery complicated by gestational hypertension (new hypertension diagnosed after 20 weeks without proteinuria), preeclampsia (high blood pressure and proteinuria diagnosed after 20 weeks gestation, or meeting other American College of Obstetrics and Gynecology diagnostic criteria consistent with preeclampsia), preterm delivery (32–36 6/7 weeks), delivery of an SGA neonate (weight <10th percentile for gestational age), and/or gestational diabetes (defined as a 3-hour 100-g oral glucose tolerance test result meeting Carpenter-Coustan criteria [45] or by medical record documented clinician diagnosis). Women were eligible regardless of the number of previous pregnancies. We identified participants by diagnosis codes, and pregnancy complications were confirmed via chart review by the study physician. Recruitment took place during prenatal or postpartum clinic visits, or after delivery at the University of Colorado Hospital on the University of Colorado Anschutz Medical Campus in Aurora, Colorado. Women were required to have access to an iPhone or iPod (Apple Inc, California) (iOS 5 or higher) because at the time of the study the FAB app was only available for iOS platforms. We excluded women with a history of preexisting diabetes, cancer, cardiovascular disease, or other major chronic illness, or a history of bariatric surgery, who delivered before 32 weeks of gestation, or who experienced

net weight loss during pregnancy. We also excluded women taking medications known to affect body weight, women planning to participate in commercial weight loss programs or planning bariatric surgery, and women unable to read eighth grade-level English. The Colorado Multiple Institutional Review Board at the University of Colorado approved the study (17–0045) on April 26, 2017, and all participants gave written informed consent.

## Study visits

We asked women to come for baseline visits at the Clinical and Translational Research Center (CTRC) at the University of Colorado at ~6 weeks postpartum. Six weeks postpartum was selected because this is the time of the typical maternal postpartum visit, including the oral glucose tolerance test for women with a recent pregnancy complicated by GDM. In addition, previous studies have shown that this is a reasonable time to begin a postpartum lifestyle intervention [24, 46, 47]. At the conclusion of the visit, we randomized women in a 2:1 ratio to the FAB mHealth Intervention group or to the Text4baby active control group using a permuted block scheme with randomly varying block sizes. Twice as many participants were randomized to the FAB program to maximize the amount of acceptability data collected from users of the program. A statistician not otherwise involved in the study prepared sealed sequentially numbered envelopes containing group assignment, and clinical research staff opened these at the end of the baseline study visit. Due to the nature of the study, neither participants nor all study staff were blinded to randomization group, but participants were blinded to the study hypotheses and whenever possible individuals who took outcome measurements were blinded to the randomization assignment of participants. The investigators including the study statistician remained blinded throughout the study. Following the baseline visit we asked participants to come to the CTRC for subsequent study visits at 6 and 12 months postpartum.

## Measures

All measures were collected at baseline, 6 month, and 12 month study visits unless otherwise noted. At each visit trained staff measured body weight twice wearing light clothing, and weights were averaged (SECA 360) and height was measured by stadiometer (SECA). We used kg/m2 to determine BMI. Trained staff also measured waist circumference. Fasting blood samples were collected to measure secondary outcomes to measure changes in cardiometabolic risk factors, including glucose, insulin, hemoglobin A1C (HbA1c), lipid profiles, adiponectin, and hsCRP. At the 6 month and 12 month visits we measured urine human chorionic gonadotropin to ensure that participants were not pregnant and serum TSH to detect abnormal thyroid function using standard assays.

## Self-reported questionnaires

Participants completed questionnaires for additional secondary outcomes including diet and physical activity changes using a validated food frequency questionnaire (2005 Block FFQ) [48], (administered via NutritionQuest), which provides an estimate of habitual intake and an adapted version of the validated Pregnancy Physical Activity Questionnaire (PPAQ), which provides a reasonably accurate measure of a broad range of physical activities [49]. Additional questionnaires included: sociodemographic, medical history, Edinburgh Postnatal Depression Scale [50] (EPDS), and breastfeeding status. Participants completed questionnaires using Research Electronic Data Capture (REDCap), a secure, HIPAA compliant web application for data collection.

## Outcomes

The two primary outcomes were change in measured body weight at 12 months from 1) first postpartum measured weight and 2) self-reported prepregnancy weight. We recorded self-reported prepregnancy weight at enrollment, either during pregnancy or after delivery and before the randomization study visit. We reviewed medical records to ascertain gestational age at delivery, and mode of delivery. We used the pregnancy weight recorded in the anesthesia record within 2 days before delivery or the last recorded prenatal weight within 10 days of delivery to calculate gestational weight gain. We used measured height with self-reported pre-pregnancy weight to calculate prepregnancy BMI. For participants who did not attend a follow-up study visit in person, we extracted clinically measured weights from the medical record.

## The Fit After Baby intervention program

Participants randomized to the FABi program were given a Fitbit and a body weight scale at the baseline study visit and shown how to use these, and then shown how to download the FAB mobile app. The FAB intervention consisted of a 12-week intensive phase with daily content centered around weekly themes, and tracking of diet, physical activity, and weight. Participants received daily notifications prompting them to open the app, and the app delivered interactive content requiring 3–10 minutes per day over the 12-week period, including quizzes, physical activity suggestions and yoga poses, recipes, inspirational quotes, and self-efficacy strategies. Participants were given a weight loss goal of returning to pre-pregnancy weight and losing additional weight up to 7% if still overweight/obese. Physical activity data from provided Fitbits were passively transmitted through wireless/Bluetooth connections, and participants had the option to manually enter physical activity into the app as well. Participants were asked in the app to gradually increase PA by 1000 steps/day each week, until they reached 10,000 steps/day, with an additional goal to increase PA by 10 minutes per day until they reached 45–60 minutes per day (the amount recommended for weight loss maintenance) [51]. Participants could earn points by opening app content, contacting the lifestyle coach, engaging in physical activity, tracking diet, setting goals, and entering weights. Points earned counted towards four levels of a "Health Warrior badge," and participants reaching each level received a small gift card via email. Completing at least 75% of all app activities would allow participants to reach the highest level.

## Lifestyle coaching

The lifestyle coach was a registered dietitian trained in motivational interviewing with previous lifestyle coaching experience during the initial FAB pilot study. Participants were also asked to communicate with the lifestyle coach via the app, or by text or phone at least weekly for the first 12 weeks, and monthly thereafter. When possible, the first lifestyle coach session was conducted over video chat to promote a personal connection. The coach viewed progress of participants weekly during the 12-week program using a dedicated FAB portal via a coaching app where she could track diet, physical activity, weight, and responses to questions. This allowed her to provide individualized coaching towards graded goals, including identifying barriers to meeting goals and strategies to overcome barriers. The study physician, trained in patient-centered counseling, periodically reviewed de-identified email and text interactions, as well as notes from phone calls, to ensure fidelity to the study. She and the coach met monthly to review this content.

### Engagement and usage data

We collected data on use of the app, including the number of days the app was opened, which content was opened, steps and minutes of physical activity, days activity trackers were worn, number of coaching interactions, and Health Warrior points accumulated. Usage data were collected in BigQuery (Google).

### Text4baby control group

Subjects randomized to the control group were shown how to download the free publicly available app Text4baby [52]. The Text4baby app delivers 2–4 free text messages per week from the Text4baby program, a nonprofit maternal child health program providing information including baby care and resources for women tailored to their number of weeks postpartum. Since Text4baby did not emphasize weight loss it served as an active control [53]. If women were already enrolled in Text4baby at the time of randomization they were asked to continue with the program through 12 months postpartum.

### Sample size determination

Based on our previous study [54], our original sample size calculation was to have 54 participants in the intervention group and 27 subjects in the control group, such that an ANCOVA controlling for baseline weight using an intent-to-treat analysis would have at least 80% power to detect a 4.2 kg difference between groups in 12-month weight change.

### Statistical analysis

We compared baseline characteristics with Pearson's chi-square or Fisher's exact tests for categorical variables and t tests for continuous variables. Participants were categorized based on treatment allocation in an intent-to-treat fashion for analyses. We compared differences between groups over time for weight, and BMIusing mixed-effects models, which allow for missing outcome data. Models included a random intercept and a compound symmetric covariance matrix and were adjusted for baseline values. Women who became pregnant were censored at the time of the event. We also estimated models that adjusted for gestational weight gain and breastfeeding. We examined changes in dietary intake over time using similar models, adjusted for kilocalorie intake at each timepoint when appropriate, and adjusted for baseline values. We used similar models to examine changes over time in physical activity and measures of cardiometabolic risk. We used a linear regression model to determine whether points earned in the program predicted weight loss. The COVID-19 pandemic interrupted data collection for the primary endpoint (weight at 12 months from baseline), and impacted diet, PA, and body weight for many participants. The University of Colorado Anschutz Medical Campus closed down clinical research visits for four months, and when operations resumed many women were unwilling to attend in-person study visits. As a sensitivity analysis, we conducted analyses excluding women who were still enrolled at the onset of the COVID pandemic in March 2020, given the well documented impact of the COVID-19 pandemic on lifestyle behaviors [55, 56]. We performed analyses using SAS version 9.4 and JMP Pro 14 (SAS Institute, Cary NC). App usage data were analyzed using BigQuery (Google) and Tableau (Mountain View, CA).

## Results

Participants were recruited from September 4, 2017 through October 7, 2019, and the study was completed August 13, 2020. The consort diagram in Fig 1 includes study enrollment

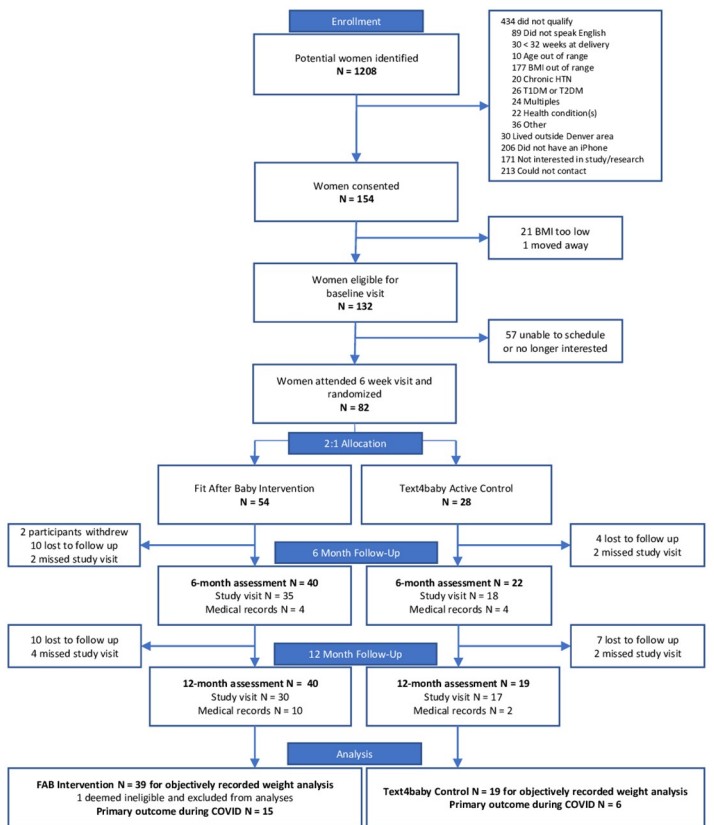

**Fig 1. Screening, recruitment, and follow-up for the Fit After Baby trial.**

and participation details. Of 1208 women identified as potentially eligible, 995 were screened for the study, 325 met initial eligibility criteria, and 154 consented to participate. Of these, 82 (53%) attended a baseline visit and were randomized 2:1 to FAB (n = 54) or the T4B program (n = 28) at 6 weeks postpartum. After removing one participant who should not have been randomized because she did not meet criteria for participation, 81 women were included for the final intent-to-treat analysis. The final follow-up visit occurred August 13, 2020. Participants were 31 (SD ±5.4) years old on average and 77% White and 15% African-American, with 24% identifying as Hispanic. Overall 53% were college graduates, 50% were primiparous, and 34% were enrolled in the Women, Infants and Children (WIC) program for low income women. The most common pregnancy complication was gestational hypertension, and 34% of participants had two or more pregnancy complications. Overall there were no significant differences in baseline characteristics between the two groups, with the exception of breastfeeding, which was significantly higher in the Text4baby group (85% vs. 62%) (Table 1). No adverse events or unintended harms occurred throughout the study.

## Primary outcome

Among all women randomized into the study, including those impacted by the COVID-19 pandemic, weight change at 12 months from baseline was -2.8 kg [95% CI -4.2, -1.4] compared to a loss of -1.8 kg [-3.8, 0.3] in the T4B group (p = 0.42 for the difference between groups). In

**Table 1. Baseline characteristics in the entire cohort and the cohort unaffected by the COVID-19 pandemic.**

| Characteristic | All Participants (N = 81) | | Participants Unaffected by COVID (N = 60) | |
|---|---|---|---|---|
| | Intervention (n = 53) (Fit After Baby) | Control (n = 28) (Text4Baby) | Intervention (n = 38) (Fit After Baby) | Control (n = 22) (Text4Baby) |
| Age, mean (SD) | 30.8 (5.5) | 31.6 (5.2) | 30.2 (5.5) | 31.7 (5.5) |
| Race: | | | | |
| White, N (%) | 42 (79%) | 20 (71%) | 31 (82%) | 17 (77%) |
| Black, N (%) | 8 (15%) | 4 (14%) | 5 (13%) | 4 (18%) |
| Asian, N (%) | 3 (6%) | 4 (14%) | 2 (5%) | 1 (4%) |
| Hispanic/Latina, N (%) | 12 (23%) | 7 (25%) | 9 (24%) | 7 (32%) |
| Education level attained: | | | | |
| Some or all of high school, N (%) | 8 (15%) | 3 (11%) | 6 (16%) | 2 (9%) |
| Some college, N (%) | 17 (32%) | 10 (36%) | 13 (34%) | 9 (41%) |
| College graduate, N (%) | 28 (53%) | 15 (54%) | 19 (50%) | 11 (50%) |
| Annual household income: | | | | |
| <$35,000, N (%) | 11 (22%) | 4 (17%) | 7 (20%) | 4 (21%) |
| $35,000–<$75,000, N (%) | 11 (22%) | 5 (21%) | 9 (26%) | 4 (21%) |
| ≥$75,000, N (%) | 28 (56%) | 15 (63%) | 19 (54%) | 11 (58%) |
| Enrolled in WIC, N (%) | 18 (38%) | 7 (27%) | 14 (41%) | 7 (33%) |
| Has a partner, N (%) | 43 (81%) | 23 (82%) | 31 (82%) | 17 (77%) |
| Primiparous, N (%) | 26 (50%) | 14 (50%) | 20 (53%) | 10 (45%) |
| Pre-pregnancy weight (kg), mean (SD) | 81.7 (15.3) | 83.6 (17.6) | 81.5 (16.8) | 84.9 (17.0) |
| Pre-pregnancy BMI (kg/m$^2$), mean (SD) | 30.3 (6.0) | 30.8 (5.2) | 30.2 (6.3) | 31.3 (5.3) |
| Gestational weight gain (kg), mean (SD) | 15.3 (8.4) | 12.2 (7.3) | 15.4 (8.9) | 12.4 (7.2) |
| Pregnancy condition: | | | | |
| Gestational diabetes, N (%) | 8 (15%) | 9 (32%) | 4 (11%) | 6 (27%) |
| Preeclampsia, N (%) | 22 (42%) | 6 (21%) | 14 (37%) | 5 (23%) |
| Gestational hypertension, N (%) | 25 (47%) | 11 (39%) | 20 (53%) | 9 (41%) |
| Pre-term delivery (32–37 weeks), N (%) | 11 (21%) | 3 (11%) | 9 (24%) | 3 (14%) |
| Small-for-gestational age (<10%ile), N (%) | 13 (25%) | 7 (25%) | 9 (24%) | 6 (27%) |
| More than one pregnancy condition, N (%) | 21 (40%) | 7 (25%) | 14 (37%) | 6 (27%) |
| Cesarean delivery, N (%) | 21 (40%) | 11 (39%) | 14 (37%) | 9 (41%) |
| Weeks postpartum at baseline visit, median (range) | 7.7 (2.7) | 8.0 (2.4) | 7.6 (2.7) | 8.1 (2.6) |
| Baseline weight (kg), mean (SD) | 87.5 (14.6) | 87.5(15.6) | 87.0 (16.0) | 89.4 (14.4) |
| Baseline BMI (kg/m$^2$), mean (SD) | 32.3 (5.2) | 32.4 (4.6) | 32.3 (5.4) | 33.1 (4.5) |
| Breastfeeding at baseline visit, n (%) | 33 (62%) | 24 (86%) | 26 (71%) | 18 (82%) |
| Depressive symptoms (EPDS ≥9) | 11 (22%) | 3 (11%) | 7 (19%) | 2 (9%) |

60 women who reached 12 months postpartum before the onset of the COVID-19 pandemic, women randomized to FAB lost 4.3 kg [-6.0, -2.6] compared to loss in the control group of 1.3 kg [-3.7, +1.1] (p = 0.0451 for difference between groups). (Table 2, Fig 2). Twenty-one participants had not reached the primary endpoint by the onset of the COVID pandemic. Of these the 15 randomized to FAB had a mean weight increase of 0.2 kg [-2.2, +2.6] compared to a mean weight loss of 3.2kg [-7.0, +0.7] in the T4B group. Adjustment for breastfeeding and gestational weight gain did not substantially change findings. A comparison of baseline characteristics and randomization assignments of participants with and without missing data showed no significant differences.

**Table 2. Change in weight and BMI from baseline and from postpartum weight in the entire cohort and the cohort unaffected by the COVID-19 pandemic*.**

| | All Participants | | | Participants unaffected by COVID | | |
|---|---|---|---|---|---|---|
| | Mean weight change (95%CI) | | | Mean weight change (95%CI) | | |
| Variable | Mean (SD) at Baseline | Month 6[†] | Month 12[††] | Mean (SD) at Baseline | Month 6[§] | Month 12[§§] |
| Control Weight (kg) | 87.5 (15.6) | -0.5 (-2.5 to 1.4) (p = 0.60) | -1.8 (-3.8 to 0.3) (p = 0.09) | 89.4 (14.4) | -0.4 (-2.6 to 1.9) (p = 0.75) | -1.3 (-3.7 to 1.1) (p = 0.30) |
| Intervention Weight (kg) | 87.5 (14.6) | -1.6 (-3.1 to -0.1) **(p = 0.0359)** | -2.8 (-4.2 to -1.4) **(p = 0.0002)** | 87.0 (16.0) | -2.3 (-4.0 to -0.6) **(p = 0.0088)** | -4.3 (-6.0 to -2.6) **(p < 0.0001)** |
| Mean difference between groups (kg), (p value) | | -1.1 (-3.5 to 1.4) (p = 0.40) | -1.0 (-3.5 to 1.5) (p = 0.42) | | -2.0 (-4.7 to 0.8) (p = 0.17) | -3.0 (-5.9 to -0.1) **(p = 0.0451)** |
| | BMI Change | | | BMI Change | | |
| | Mean (SD) at Baseline | Month 6[†] | Month 12[††] | Mean (SD) at Baseline | Month 6[§] | Month 12[§§] |
| Control BMI (kg/m$^2$) | 32.4 (4.6) | - 0.1 (-0.8 to 0.6) (p = 0.76) | -0.5 (-1.3 to 0.3) (p = 0.20) | 33.1 (4.5) | -0.0 (-0.9 to 0.8) (p = 0.93) | -0.5 (-1.4 to 0.4) (p = 0.28) |
| Intervention BMI (kg/m$^2$) | 32.3 (5.2) | -0.5 (-1.1 to 0.0) (p = 0.0589) | -1.3 (-1.8 to -0.7) **(p < 0.0001)** | 32.3 (5.4) | -0.8 (-1.5 to -0.2) **(p = 0.0148)** | -1.6 (-2.3 to -0.9) **(p < 0.0001)** |
| Mean difference between groups (kg/m$^2$) | | -0.4 (-1.3 to 0.5) (p = 0.37) | -0.8 (-1.7 to 0.2) (p = 0.13) | | -0.8 (-1.8 to 0.3) (p = 0.16) | -1.1 (-2.2 to 0) (p = 0.0535) |
| | Weight change from pre-pregnancy weight, mean (95%CI) | | | Weight change from pre-pregnancy weight, mean (95%CI) | | |
| | | Month 6[†] | Month 12[††] | | Month 6[§] | Month 12[§§] |
| Control (kg) | | 2.9 (0.2 to 5.6) **(p = 0.0342)** | 1.5 (-1.3 to 4.3) (p = 0.30) | | 3.7 (0.6 to 6.9) **(p = 0.0211)** | 2.7 (-0.7 to 6.0) (p = 0.12) |
| Intervention (kg) | | 3.8 (1.8 to 5.8) **(p = 0.0002)** | 2.5 (0.6 to 4.5) **(p = 0.0115)** | | 2.4 (0.0 to 4.8) **(p = 0.0485)** | 0.57 (-1.8 to 3.0) (p = 0.64) |
| Mean difference between groups | | 0.9 (-2.4 to 4.3) (p = 0.58) | 1.1 (-2.3 to 4.4) (p = 0.54) | | -1.3 (-5.2 to 2.6) (p = 0.51) | -2.1 (-6.2 to 2.0) (p = 0.31) |

*Models adjusted for baseline weight/BMI.

[†]There were data from 61 participants at the 6-month time point, but for the purposes of analysis, the model predicted data for all 81 women

[††]There were data from 51 participants at the 12-month time point, but for the purposes of analysis, the model predicted data for all 81 women

[§]There were data from 48 participants at the 6-month time point, but for the purposes of analysis, the model predicted data for all 60 women unaffected by COVID

[§§]There were data from 45 participants at the 12-month time point, but for the purposes of analysis, the model predicted data for all 60 women unaffected by COVID

## Changes in diet, physical activity, and measures of cardiometabolic risk

Table 3 shows the baseline values for secondary outcomes of diet, physical activity and measures of cardiometabolic risk among the 60 participants unaffected by the COVID pandemic. Total energy intake, glycemic load, and percentage from saturated fat were different between groups at baseline. Table 4 demonstrates changes in diet among the 60 participants unaffected by the COVID pandemic. There were no significant differences in diet change between groups. Within the FAB intervention group, significant decreases were observed in overall kilocalorie intake, % kilocalories from carbohydrates, glycemic load, and % of kilocalories from sweets at 6 and 12 months, as well as a significant increase in vegetable servings at 6 months. A decrease in the percent of calories from sweets and increase in vegetable servings were also significant in the control group at 12 months (Table 4). Both groups significantly decreased their sedentary activity during the study period but there was no difference between groups. Both groups decreased their moderate and light physical activity as well (Table 5).

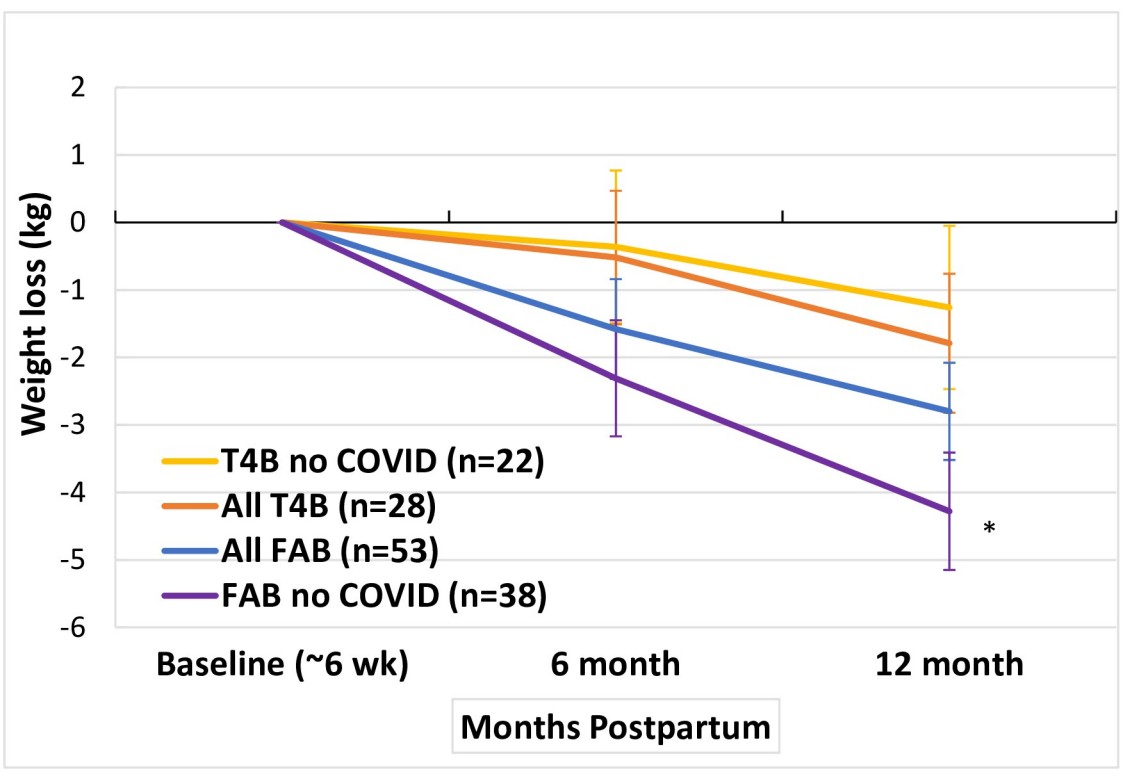

**Fig 2. Postpartum weight loss in the Fit After Baby trial.** *Significantly different when compared to T4B no COVID.

## Changes in cardiometabolic risk measures

Table 6 shows the changes in cardiometabolic risk indices. These were not different between groups. However, within the intervention group, among those not impacted by COVID, there was a significant decrease in LDL at 6 and 12 months from baseline, a significant increase in adiponectin at 6 and 12 months from baseline, and a significant decrease in HDL at 6 months from baseline. Within the control group there was a significant decrease in LDL at 6 months from baseline. There was no significant difference in change in waist circumference between groups, however within the intervention group there was a significant decrease in waist circumference from baseline by 5.3 and 7.1 cm, at 6 and 12 months respectively, and a significant decrease of 4.0 cm at 12 months in the control group (Table 6).

## Acceptability and user engagement

Participants logged into the app a median of 61% of all days over the first 12 weeks (51/84, IQR 25,71), wore activity trackers a median of 79% of days (66/84, IQR 43,84), entered their weight a median of 17 times (IQR 11,24), and completed weekly coach check-ins 5.5/12 (IQR 4,9) weeks. The majority of participants (45/53, 85%) reached the Bronze level of the Health Warrior badge, signifying 18.5% of all app related tasks completed. 41/53 (77%) reached Silver, 36/53 (68%) reached Gold, and 34/63 (64%) reached Platinum, signifying 37.5%, 56% and 75% of all app-related tasks completed, respectively. Accumulating total points in the app was significantly associated with greater weight loss at 6 months (p < .05). A greater number of interactions with the coach, more days wearing an activity tracker, and greater accumulation of reward points all predicted weight loss at 12 months (all p<0.05).

**Table 3. Baseline data for diet, physical activity, and cardiometabolic risk factors among 60 participants unaffected by the COVID pandemic.**

| Characteristic | Intervention | Control | P-Value |
|---|---|---|---|
| | Fit After Baby (n = 38) | Text4Baby (n = 22) | |
| **Dietary Intake** | | | |
| Kcal, median(IQR) | 1824.0 (1463.5,2134.8) | 2255.9 (1700.2,2651.2) | **0.0109** |
| Percent of Kcal from Carbohydrates, mean(SD) | 44.9 (8.4) | 44.0 (5.9) | 0.6701 |
| Glycemic Load, median(IQR) | 89.6 (73.9, 116.4) | 112.3 (89.3, 137.5) | **0.0242** |
| Saturated Fat (g), median(IQR) | 25.4 (23.1, 30.7) | 30.1 (26.1, 38.3) | **0.0122** |
| Dietary Fiber (g), median(IQR) | 17.2 (15.6, 22.2) | 20.3 (16.5, 25.2) | 0.0801 |
| Percent of Kcal from Sweets, mean(SD) | 15.2 (9.4) | 14.6 (8.1) | 0.8070 |
| Vegetable Servings, mean(SD) | 2.8 (1.5) | 2.6 (1.9) | 0.6712 |
| Fruit Servings, mean(SD) | 1.4 (1.0) | 1.2 (0.8) | 0.5899 |
| Whole Grain Servings, median(IQR) | 0.3 (0.1, 0.8) | 0.6 (0.5, 0.9) | 0.3713 |
| **Physical Activity** | | | |
| Sedentary Hours per Week, mean(SD) | 47.3 (17.9) | 46.7 (24.1) | 0.9214 |
| Moderate Activity Hours per Week, mean(SD) | 50.6 (28.5) | 47.9 (23.3) | 0.7085 |
| Light Activity Hours per Week, mean(SD) | 57.9 (20.8) | 57.0 (19.4) | 0.8773 |
| **Lab Assessment** | | | |
| Fasting Glucose, mean(SD) | 84.8 (7.9) | 87.1 (9.5) | 0.3583 |
| Fasting Triglycerides, median(IQR) | 87.0 (58.0, 148.0) | 83.0 (69.0, 108.0) | 0.2849 |
| Low Density Lipoprotein (LDL), mean(SD) | 114.5 (30.2) | 114.8 (27.7) | 0.9766 |
| High Density Lipoprotein (HDL), mean(SD) | 51.9 (11.4) | 49.9 (10.0) | 0.4876 |
| C Reactive Protein (CRP), median(IQR) | 2.5 (1.5, 5.8) | 4.5 (2.6, 6.5) | 0.1749 |
| Adiponectin, mean(SD) | 9.2 (4.6) | 8.8 (4.1) | 0.7555 |
| Waist Circumference, mean(SD) | 107.4 (13.6) | 108.1 (9.5) | 0.8297 |
| Systolic BP, mean(SD) | 115.0 (8.5) | 114.1 (10.9) | 0.7335 |
| Diastolic BP, mean(SD) | 74.6 (7.9) | 71.4 (10.0) | 0.1794 |
| HOMA-IR, mean(SD) | 3.5 (0.3) | 3.6 (0.4) | 0.1552 |
| Hemoglobin A1c, mean(SD) | 5.2 (0.3) | 5.3 (0.2) | 0.2952 |

## Discussion

In this study we did not find a significant difference between the FAB and Text4baby groups in postpartum weight loss. The latter part of this study was conducted during the early days of the COVID-19 pandemic, which likely impacted the results. In our sensitivity analysis looking at participants unaffected by the COVID pandemic, those randomized to the Fit After Baby program lost significantly more weight than those randomized to the Text4baby program (active control). We demonstrated promising engagement with the intervention and notably the level of engagement with coaching and gamification within the app predicted weight loss. Given the many constraints on face-to-face participation, employing an mHealth strategy is an innovative approach to reach this high-risk population.

Twenty-five percent of our participants were still enrolled in the study at the onset of the COVID-19 pandemic, which impacted our results. Other studies have been impacted by the effect of the pandemic on diet, exercise, and weight loss [55, 56], which, particularly early in the pandemic, likely overwhelmed the impact of a lower intensity intervention. The lack of significant difference between groups may be due to the higher than typical weight loss in the control group, particularly after the start of the COVID pandemic. Since the active part of the FAB program was the first 12 weeks, the COVID pandemic began when FAB participants were no longer in the active phase of the intervention. Women in the Text4baby group,

**Table 4. Changes in diet over time among participants unaffected by the COVID-19 pandemic[#].**

| Group Assignment | Variable | |
|---|---|---|
| | Kilocalories (Kcal) | |
| | Month 6[§] | Month 12[§§] |
| Control (Text4Baby) | -175.3 (-722.2 to 371.5) (p = 0.52) | -262.3 (-809.2 to 284.5) (p = 0.34) |
| Intervention (Fit After Baby) | -718.2 (-1107.5 to -328.9) **(p = 0.0004)** | -784.1 (-1186.7 to -381.5) **(p = 0.0002)** |
| Mean difference between groups | -542.9 (-1203.3 to 117.5) (p = 0.11) | -521.8 (-1189.8 to 146.2) (p = 0.13) |
| | % of Kcal from Carbohydrates | |
| | Month 6[§] | Month 12[§§] |
| Control (Text4Baby) | -2.8 (-6.5 to 0.8) (p = 0.13) | -3.3 (-7.0 to 0.4) (p = 0.08) |
| Intervention (Fit After Baby) | -2.9 (-5.4 to -0.3) **(p = 0.0310)** | -6.5 (-9.2 to -3.8) **(p < 0.0001)** |
| Mean difference between groups | 0.0 (-4.4 to 4.4) (p = 0.99) | -3.2 (-7.7 to 1.3) (p = 0.16) |
| | Glycemic Load (% Change) | |
| | Month 6[§] | Month 12[§§] |
| Control (Text4Baby) | -19.9 (-38.3 to 4.0) (p = 0.09) | -24.1 (-41.6 to -1.5) **(p = 0.0384)** |
| Intervention (Fit After Baby) | -35.9 (-46.7 to -23.0) **(p = < .0001)** | -44.2 (-54.0 to -32.5) **(p = < .0001)** |
| Mean difference between groups | -20.0 (-41.6 to 9.5) (p = 0.17) | -26.5 (-46.5 to 1.0) (p = 0.06) |
| | Saturated Fat* (% Change) | |
| | Month 6[§] | Month 12[§§] |
| Control (Text4Baby) | -0.24 (-9.2 to 9.6) (p = 0.96) | 3.5 (-5.8 to 13.8) (p = 0.47) |
| Intervention (Fit After Baby) | -5.0 (-11.5 to 2.0) (p = 0.16) | 5.6 (-2.0 to 13.8) (p = 0.15) |
| Mean difference between groups | -4.8 (-15.0 to 6.6) (p = 0.40) | 2.0 (-9.0 to 14.4) (p = 0.73) |
| | Fiber* (% Change) | |
| | Month 6[§] | Month 12[§§] |
| Control (Text4Baby) | -3.6 (-15.5 to 10.0) (p = 0.58) | -8.4 (-19.8 to 4.5) (p = 0.19) |
| Intervention (Fit After Baby) | -1.9 (-11.1 to 8.4) (p = 0.71) | -9.0 (-17.9 to 1.0) (p = 0.08) |
| Mean difference between groups | 1.8 (-13.2 to 19.4) (p = 0.83) | -0.6 (-15.4 to 16.8) (p = 0.94) |
| | % of Kcal from Sweets | |
| | Month 6[§] | Month 12[§§] |
| Control (Text4Baby) | -2.3 (-6.0 to 1.3) (p = 0.21) | -6.0 (-9.6 to -2.4) **(p = 0.0015)** |
| Intervention (Fit After Baby) | -5.1 (-7.7 to -2.5) **(p = 0.0002)** | -6.7 (-9.4 to -4.1) **(p < .0001)** |
| Mean difference between groups | -2.7 (-7.1 to 1.6) (p = 0.22) | -0.74 (-5.2 to 3.7) (p = 0.74) |
| | Vegetable Servings (per day) | |
| | Month 6[§] | Month 12[§§] |
| Control (Text4Baby) | 1.0 (0.1 to 1.8) **(p = 0.0253)** | 0.52 (-0.3 to 1.4) (p = 0.22) |
| Intervention (Fit After Baby) | 0.63 (0.0 to 1.2) **(p = 0.0381)** | 0.23 (-0.4 to 0.9) (p = 0.47) |
| Mean difference between groups | -0.33 (-1.4 to 0.7) (p = 0.52) | -0.3 (-1.3 to 0.7) (p = 0.57) |
| | Fruit Servings (per day) | |
| | Month 6[§] | Month 12[§§] |
| Control (Text4Baby) | 0.2 (-0.2 to 0.6) (p = 0.37) | 0.2 (-0.3 to 0.6) (0.46) |
| Intervention (Fit After Baby) | 0.0 (-0.3 to 0.3) (p = 0.94) | -0.3 (-0.6 to 0.0) (p = 0.09) |
| Mean difference between groups | -0.2 (-0.7 to 0.3) (p = 0.44) | -0.4 (-0.9 to 0.1) (p = 0.11) |
| | Whole Grains* (% change) | |
| | Month 6[§] | Month 12[§§] |
| Control (Text4Baby) | -7.4 (-54.3 to 87.5) (p = 0.83) | 52.7 (-24.7 to 209.9) (p = 0.24) |
| Intervention (Fit After Baby) | -20.0 (-52.5 to 34.7) (p = 0.40) | -18.7 (-52.8 to 40.1) (p = 0.45) |

(*Continued*)

**Table 4.** (Continued)

| Group Assignment | Variable | |
|---|---|---|
| Mean difference between groups | -13.6 (-63.3 to 103.4) (p = 0.74) | -46.7 (-77.6 to 26.6) (p = 0.16) |

[#]All models adjusted for baseline values

[*]Adjusted for total kilocalorie intake. All log transformed dependent variables have been back transformed with formular of $(\exp(\beta)- 1)*100$.

[†]There were data from 45 participants at the 6-month time point, but for the purposes of analysis, the model predicted data for all 81 women

[††]There were data from 40 participants at the 12-month time point, but for the purposes of analysis, the model predicted data for all 81 women

[$]There were data from 34 participants at the 6-month time point, but for the purposes of analysis, the model predicted data for all 60 women unaffected by COVID

[§§]There were data from 36 participants at the 12-month time point, but for the purposes of analysis, the model predicted data for all 60 women unaffected by COVID

however, were still receiving 3–4 texts per week, which may have helped with a sense of connection and may have promoted behaviors such as continuation of breastfeeding leading to increased weight loss in the Text4baby group. In addition, women in the control group were significantly more likely to be breastfeeding at baseline, which may have influenced their weight loss. We show a significant difference in weight loss of 3.0 kg in the intervention group compared to the control group among women completing the study before the onset of the COVID-19 pandemic. The difference in weight loss achieved prior to the pandemic with this

**Table 5.** Changes in measures of physical activity among participants unaffected by the COVID-19 pandemic[*][#].

| Sedentary Hours Per Week | Month 6[$] | Month 12[§§] |
|---|---|---|
| Control (Text4Baby) | -19.0 (-30.5 to -7.5) (**p = 0.001**) | -16.5 (-28.0 to -5.0) (**p = 0.005**) |
| Intervention (Fit After Baby) | -19.4 (-28.2 to -10.7) (**p < 0.0001**) | -19.6 (-28.3 to -10.8) (**p < 0.0001**) |
| Mean difference between groups | -0.04 (p = 0.96) | -3.1 (p = 0.67) |
| Moderate Activity Hours Per Week | Month 6[$] | Month 12[§§] |
| Control (Text4Baby) | -18.3 (-32.7 to -4.0) (**p = 0.01**) | -12.8 (-27.1 to 1.5) (p = 0.08) |
| Intervention (Fit After Baby) | -16.6 (-27.5 to -5.7) (**p = 0.003**) | -18.8 (-29.7 to -5.7) (**p = 0.001**) |
| Mean difference between groups | 1.70 (p = 0.85) | -6.0 (p = 0.51) |
| Light Activity Hours Per Week | Month 6[$] | Month 12[§§] |
| Control (Text4Baby) | -19.0 (-31.4 to -6.5) (**p = 0.003**) | -22.3 (-34.8 to -9.8) (**p = 0.0006**) |
| Intervention (Fit After Baby) | -22.5 (-32.0 to 13.0) (**p < 0.0001**) | -26.6 (-36.0 to -17.1) (**p < 0.0001**) |
| Mean difference between groups | -3.57 (p = 0.65) | -4.25 (p = 0.59) |

[#]All models adjusted for baseline values

[*]Not enough data for vigorous activity for participants unaffected by COVID

[†]There were data from 45 participants at the 6-month time point, but for the purposes of analysis, the model predicted data for all 81 women

[††]There were data from 40 participants at the 12-month time point, but for the purposes of analysis, the model predicted data for all 81 women

[$]There were data from 34 participants at the 6-month time point, but for the purposes of analysis, the model predicted data for all 60 women unaffected by COVID

[§§]There were data from 36 participants at the 12-month time point, but for the purposes of analysis, the model predicted data for all 60 women unaffected by COVID

**Table 6. Changes in measures of cardiometabolic risk among participants unaffected by the COVID-19 pandemic[#].**

| Group | Variable | |
|---|---|---|
| | Fasting Glucose* (% change) | |
| | Month 6[§] | Month 12[§§] |
| Control (Text4Baby) | 2.0 (-2.5 to 6.7) (p = 0.39) | 6.2 (1.4 to 11.2) **(p = 0.0112)** |
| Intervention (Fit After Baby) | -0.2 (-3.1 to 2.9) (p = 0.91) | 2.6 (-0.5 to 5.9) (p = 0.10) |
| Mean difference between groups | -2.1 (-7.2 to 3.3) (p = 0.44) | -3.4 (-8.5 to 2.0) (p = 0.22) |
| | Fasting Triglycerides* (% change) | |
| | Month 6[§] | Month 12[§§] |
| Control (Text4Baby) | -11.2 (-26.2 to 6.9) (p = 0.21) | -0.1 (-17.0 to 20.3) (p = 0.99) |
| Intervention (Fit After Baby) | -9.8 (-20.9 to 3.0) (p = 0.13) | -10.0 (-21.4 to 3.0) (p = 0.12) |
| Mean difference between groups | 1.6 (-18.8 to 27.0) (p = 0.89) | -10.0 (-28.1 to 12.8) (p = 0.37) |
| | Low Density Lipoprotein (LDL) mg/dL | |
| | Month 6[§] | Month 12[§§] |
| Control (Text4Baby) | -16.4 (-26.0 to -6.7) **(p = 0.0011)** | -8.8 (-18.4 to 0.8) (p = 0.07) |
| Intervention (Fit After Baby) | -15.6 (-22.4 to -8.8) **(p < .0001)** | -14.7 (-21.6 to -7.7) **(p = 0.0001)** |
| Mean difference between groups | 0.82 (-10.8 to 12.4) (p = 0.89) | -5.9 (-17.6 to 5.8) (p = 0.33) |
| | High-density Lipoprotein (HDL) (mg/dL) | |
| | Month 6[§] | Month 12[§§] |
| Control (Text4Baby) | -2.3 (-6.2 to 1.7) (p = 0.25) | -3.3 (-7.3 to 0.6) (p = 0.09) |
| Intervention (Fit After Baby) | -4.1 (-6.9 to -1.3) **(p = 0.0043)** | -2.6 (-5.4 to 0.3) (p = 0.08) |
| Mean difference between groups | -1.9 (-6.6 to 2.9) (p = 0.45) | 0.8 (-4.0 to 5.6) (p = 0.74) |
| | C-Reactive Protein* (% change) | |
| | Month 6[§] | Month 12[§§] |
| Control (Text4Baby) | 8.0 (-32.9 to 73.7) (p = 0.75) | -16.6 (-48.2 to 34.1) (p = 0.45) |
| Intervention (Fit After Baby) | 33.1 (-5.1 to 86.9) (p = 0.10) | -23.1 (-45.5 to 8.3) (p = 0.13) |
| Mean difference between groups | 23.3 (-30.5 to 118.6) (p = 0.48) | -7.8 (-48.3 to 64.2) (p = 0.78) |
| | Adiponectin (ug/mL) | |
| | Month 6[§] | Month 12[§§] |
| Control (Text4Baby) | 0.3 (-1.8 to 2.4) (p = 0.78) | 0.5 (-1.6 to 2.6) (p = 0.62) |
| Intervention (Fit After Baby) | 1.9 (0.4 to 3.4) **(p = 0.0119)** | 2.5 (1.0 to 4.0) **(p = 0.0012)** |
| Mean difference between groups | 1.6 (-0.9 to 4.1) (p = 0.21) | 2.0 (-0.5 to 4.5) (p = 0.12) |
| | Waist Circumference (cm) | |
| | Month 6[§] | Month 12[§§] |
| Control (Text4Baby) | -3.0 (-6.4 to 0.3) (p = 0.08) | -4.0 (-7.3 to -0.7) **(p = 0.0168)** |
| Intervention (Fit After Baby) | -5.3 (-7.7 to -2.9) **(p < .0001)** | -7.1 (-9.5 to -4.7) **(p < .0001)** |
| Mean difference between groups | -2.3 (-6.3 to 1.8) (p = 0.27) | -3.1 (-7.1 to 0.9) (p = 0.13) |
| | Systolic Blood Pressure (mmHg) | |
| | Month 6[§] | Month 12[§§] |
| Control (Text4Baby) | 2.0 (-2.7 to 6.7) (p = 0.41) | -1.9 (-7.0 to 3.1) (p = 0.45) |
| Intervention (Fit After Baby) | 0.5 (-3.0 to 4.0) (p = 0.78) | -1.8 (-5.3 to 1.7) (p = 0.32) |
| Mean difference between groups | -1.5 (-7.2 to 4.3) (p = 0.62) | 0.2 (-5.9 to 6.3) (p = 0.95) |
| | Diastolic Blood Pressure (mmHg) | |
| | Month 6[§] | Month 12[§§] |
| Control (Text4Baby) | 0.5 (-3.7 to 4.8) (p = 0.81) | 0.9 (-3.7 to 5.5) (p = 0.70) |
| Intervention (Fit After Baby) | -0.3 (-3.5 to 2.9) (p = 0.86) | -1.7 (-4.9 to 1.4) (p = 0.28) |
| Mean difference between groups | -0.8 (-6.0 to 4.4) (p = 0.77) | -2.6 (-8.1 to 2.9) (p = 0.35) |

(*Continued*)

**Table 6.** (Continued)

| Group | Variable | |
|---|---|---|
| | HOMA-IR | |
| | Month 6[§] | Month 12[§§] |
| Control (Text4Baby) | 0.1 (-0.1 to 0.2) (p = 0.41) | 0.2 (0.0 to 0.3) (p = 0.07) |
| Intervention (Fit After Baby) | 0.0 (-0.2 to 0.1) (p = 0.54) | 0.1 (0.0 to 0.2) (p = 0.23) |
| Mean difference between groups | -0.1 (-0.3 to 0.1) (p = 0.31) | -0.1 (-0.3 to 0.1) (p = 0.38) |

[#]All models adjusted for baseline values

[*]All log transformed dependent variables have been back transformed with formula of $(exp(\beta)-1)*100$ to arrive at percentage change

[†]There were data from 45 participants at the 6-month time point, but for the purposes of analysis, the model predicted data for all 81 women

[††]There were data from 40 participants at the 12-month time point, but for the purposes of analysis, the model predicted data for all 81 women

[§]There were data from 34 participants at the 6-month time point, but for the purposes of analysis, the model predicted data for all 60 women unaffected by COVID

[§§]There were data from 36 participants at the 12-month time point, but for the purposes of analysis, the model predicted data for all 60 women unaffected by COVID

intervention is promising given the influence of the postpartum period for determining future obesity and cardiometabolic disease. Postpartum women may need ongoing and more intensive interactions with a lifestyle program to continue/maintain lifestyle behaviors. Future versions of the intervention would benefit from a longer intervention period and a more active maintenance phase.

The difference in weight loss between groups in our study, about 3 kg, is similar to the majority of studies aiming to increase postpartum weight loss. Although there is a lot of variability in postpartum weight loss, studies show that 15–27% of women have major postpartum weight retention at one year of at least 4.55 kg [28]. Nearly all women (97%) who have obesity before pregnancy will continue to be classified as such at one year [57], with 40% increasing by two or more BMI units [58]. Among women with overweight, 40–50% will move into the obesity category by 12 months postpartum [59]. A recent systematic review of 9 lifestyle intervention studies among postpartum women showing a mean weight loss of 1.7 kg [35] and meta-analysis of 46 articles showed a mean weight difference of 2.5 kg [60]. The Mothers after Gestational Diabetes in Australia (MAGDA) study enrolled 573 Australian women with previous GDM into a trial of 5 in-person and 2 telephone sessions. There was a small significant difference in body weight between groups at 12 months of 1 kg [61], and the Active Mothers Postpartum study of 450 overweight/obese postpartum women showed no significant difference in weight loss between groups at 12 months [62]. The Gestational Diabetes' Effects on Moms (GEMS) pragmatic trial in 2,280 women with GDM utilized mailings during pregnancy and delivered a Diabetes Prevention Program [34] (DPP)-derived intervention postpartum by 13 phone counseling sessions. They found a modest improvement in reaching weight goals, with a significant difference between groups at 6 months of -0.64 kg (95% CI -1.13,-0.14) but not at 12 months [63].

Two studies addressing postpartum weight retention with mobile apps also showed small effect sizes. In one study among women receiving WIC, women randomized to a personalized health intervention delivered via the "E-Moms" app did not show more weight loss, but women with high adherence to the intervention did have a significant change in weight and

percent body fat [64]. In one recent study of 200 postpartum women with a GDM history in Singapore in the Smartphone App to Restore Optimal Weight (SPAROW), those randomized to a mobile app lost 1 kg more than those randomized to control, which did not reach significance (p = .08) [32].

We demonstrated promising engagement in our study, as compared to other similar programs in postpartum women. Women in our study interacted with the app more than half of the days during the 12 week intervention and wore activity trackers more than 75% of the time. More than half of participants in the FAB program reached the highest level of points corresponding to at least 75% of all app-related tasks completed and content read. Previous studies in this population have shown varying levels of engagement. In the MAGDA study, only 10% of participants completed all sessions, and 34% attended no sessions at all. Among the 1,087 women randomized to the GEMS intervention, only 50% completed one or more telephone sessions, with just 15% completing all 13 sessions [63]. In the Fit Moms study for low-income women, participants spent about 3 hours of total time on the intervention website during the year-long intervention [65]. Although the SPAROW trial did not show significant differences in weight loss, Singaporean participants used at least one component of the mobile app for 66% of the days of the first four months, and made significant dietary changes, suggesting that mobile apps may promote better engagement than other methods for postpartum women [32]. Engagement is a key factor for intervention efficacy [66], and has been associated with increased weight loss and improvement in healthy behaviors [67, 68].

We were not powered for our secondary outcomes, but we did see some promising changes within the intervention group with respect to diet and markers of cardiometabolic risk. Decreasing saturated fat intake was one of the primary diet changes emphasized during the week of content focusing on fat. There were many other significant dietary changes within the intervention group as well, including kilocalorie intake, glycemic load, and the percent of calories from carbohydrate and added sugars. Although we did not see changes in lab values overall between groups, there were significant within group differences for LDL and adiponectin in the expected direction in the intervention group. Pregnancy is known to be associated with a more atherogenic lipid profile, and this tends to improve in the postpartum period [42, 69] in all women and to a greater degree in lactating women [70].

We did not see significant differences between randomized groups in changes in physical activity. Postpartum women rarely meet the American College of Obstetricians and Gynecologists recommendations for 150 minutes of moderate intensity physical activity per week [13], and this is even lower among women with a history of APOs [15–17]. Other studies of lifestyle interventions in postpartum women have not shown a significant difference between groups in physical activity [54, 71–73]. In addition, as has been seen in other studies, the PPAQ may overestimate the amount of physical activity in the early postpartum period, given the higher level of physical activity attributed to household activities with babies, including carrying babies around the house and pushing babies in strollers [74]. This may be why the physical activity estimates are highest in the early postpartum months in our study.

Strengths of our study include the randomized controlled design and promising engagement with the intervention when compared to similar interventions in this population. There are several limitations to our study. The COVID-19 pandemic affected retention as well as outcomes, and therefore the observed effect size was smaller than the effect size for which the study was powered for the primary weight outcome at 12 months postpartum. As a single-center trial, results may not be generalizable to other regions. We had a large proportion of eligible women decline to participate and a large number of consented women who did not come to a baseline visit. This is often seen in postpartum studies, as postpartum women face a lot of

barriers to attending in-person study visits, and this may limit generalizability in that women who do participate may be more motivated than those who do not. In addition, because the intervention was only available for participants with an iOS operating system at the time of the study, our population may have been biased towards those of higher socioeconomic status. We collected pre-pregnancy weight by self-report, which could result in recall bias; however, other studies have demonstrated a strong correlation between self-reported and clinically measured pre-pregnancy weight [75]. In addition, these self-reported data were obtained before randomization and therefore the effect of the bias on the primary outcome should be minimized. We collected physical activity and dietary data by self-report, which are not as accurate as objective measures of physical activity and diet. We had some missing data, but since the participants with and without missing data did not differ for major characteristics, this was unlikely to cause significant bias. In addition, we adjusted for baseline values in our models.

In summary, although the Fit After Baby mHealth intervention did not show a significant overall difference in postpartum weight loss as compared to the Text4baby program, those randomized to the Fit After Baby mHealth intervention who were unaffected by the COVID pandemic had significantly more weight loss than those randomized to the Text4baby program. We demonstrated substantial engagement with the intervention. An improved version of the app, with increased intensity, more interactive coaching, and a longer maintenance period, may be more effective. We will also consider whether using updated technology like smart watches and building upon gamification may be more effective for weight loss.

## Supporting information

**S1 Checklist. CONSORT 2010 checklist of information to include when reporting a randomised trial\*.**
(DOC)

**S1 File.**
(DOCX)

## Acknowledgments

We would like to thank the Fit After Baby participants. We would like to acknowledge the support from research assistants Danielle Cook, Chelsea Arent, Emily Dunn, and Jamie Siegart. We are very grateful for extensive contributions and support from Dean Hovey, Susan Gilbert, Sue Arment, and Glenn Bachmann.

## Author Contributions

**Conceptualization:** Jacinda M. Nicklas, Jennifer A. Leiferman, Sheana S. Bull, Linda A. Barbour.

**Data curation:** Jacinda M. Nicklas, Andrey Soares, Suhong Tong.

**Formal analysis:** Jacinda M. Nicklas, Laura Pyle, Andrey Soares, Suhong Tong.

**Funding acquisition:** Jacinda M. Nicklas.

**Investigation:** Jacinda M. Nicklas, Jennifer A. Leiferman, Sheana S. Bull, Linda A. Barbour.

**Methodology:** Jacinda M. Nicklas, Jennifer A. Leiferman, Sheana S. Bull, Linda A. Barbour.

**Project administration:** Jacinda M. Nicklas.

**Resources:** Jacinda M. Nicklas.

**Software:** Andrey Soares.

**Supervision:** Jacinda M. Nicklas, Sheana S. Bull, Nanette Santoro, Linda A. Barbour.

**Validation:** Jacinda M. Nicklas, Laura Pyle.

**Visualization:** Jacinda M. Nicklas.

**Writing – original draft:** Jacinda M. Nicklas.

**Writing – review & editing:** Jacinda M. Nicklas, Laura Pyle, Andrey Soares, Jennifer A. Leiferman, Sheana S. Bull, Ann E. Caldwell, Nanette Santoro, Linda A. Barbour.

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
