## [Decision Letter · Decision Letter 0]

4 Apr 2023

PONE-D-23-03981The Fit After Baby randomized controlled trial: An mHealth postpartum lifestyle intervention for women with elevated cardiometabolic riskPLOS ONE

Dear Dr. Nicklas,

Thank you for submitting your manuscript to PLOS ONE. After careful consideration, we feel that it has merit but does not fully meet PLOS ONE’s publication criteria as it currently stands. Therefore, we invite you to submit a revised version of the manuscript that addresses the points raised during the review process.

Please see the comments from three subject matter expert reviewers and one statistical reviewer below, who all appear positive about the study and its contributions. Whilst we acknowledge that this gives you a lot of reviewer comments to attend to, we feel that all reviewers have provided important insight, and we hope that by addressing these the presentation of the study will be strengthened. We look forward to receiving your revised manuscript, and please do get in touch to request an extension if you feel you need more time.

We look forward to receiving your revised manuscript.

Kind regards,

Hanna Landenmark

Staff Editor

PLOS ONE

Journal Requirements:

2. Please expand the acronym “NHLBI” (as indicated in your financial disclosure) so that it states the name of your funders in full.

"We would like to thank the Fit After Baby participants. We would like to acknowledge the support from research assistants Danielle Cook, Chelsea Arent, Emily Dunn, and Jamie Siegart. We are very grateful for extensive contributions and support from Dean Hovey, Susan Gilbert, Sue Arment, and Glenn Bachmann. This study was supported by National Institutes of Health NIH (BIRCWH K12 HD057022 and NIH NHLBI 1K23HL133604) and NIH/National Center for Advancing Translational Sciences Colorado (CTSA UL1 TR002541). This trial is registered at ClinicalTrials.gov, NCT03215173 (https://clinicaltrials.gov/ct2/show/NCT03215173). "

"Author JMN was supported by three grants: 

- NIH BIRCWH K12 HD057022, National Institutes of Health, URL: https://orwh.od.nih.gov/career-development-education/building-interdisciplinary-research-careers-in-womens-health-bircwh 

NIH NHLBI 1K23HL133604

https://www.nhlbi.nih.gov/grants-and-training/training-and-career-development/early-career

- NIH/NCATS Colorado CTSA UL1 TR002541, National Institutes of Health, National Center for Advancing Translational Sciences, URL: https://ncats.nih.gov/ctsa 

7. We note that the original protocol file you uploaded contains a confidentiality notice indicating that the protocol may not be shared publicly or be published. Please note, however, that the PLOS Editorial Policy requires that the original protocol be published alongside your manuscript in the event of acceptance. Please note that should your paper be accepted, all content including the protocol will be published under the Creative Commons Attribution (CC BY) 4.0 license, which means that it will be freely available online, and any third party is permitted to access, download, copy, distribute, and use these materials in any way, even commercially, with proper attribution.

Therefore, we ask that you please seek permission from the study sponsor or body imposing the restriction on sharing this document to publish this protocol under CC BY 4.0 if your work is accepted. We kindly ask that you upload a formal statement signed by an institutional representative clarifying whether you will be able to comply with this policy. Additionally, please upload a clean copy of the protocol with the confidentiality notice (and any copyrighted institutional logos or signatures) removed.

Reviewers' comments:

Reviewer's Responses to Questions

**Comments to the Author**

1. Is the manuscript technically sound, and do the data support the conclusions?

Reviewer #1: Partly

Reviewer #2: Yes

Reviewer #3: Yes

Reviewer #4: Yes

2. Has the statistical analysis been performed appropriately and rigorously? 

Reviewer #1: Yes

Reviewer #2: Yes

Reviewer #3: Yes

Reviewer #4: Yes

3. Have the authors made all data underlying the findings in their manuscript fully available?

Reviewer #1: No

Reviewer #2: Yes

Reviewer #3: Yes

Reviewer #4: No

4. Is the manuscript presented in an intelligible fashion and written in standard English?

Reviewer #1: No

Reviewer #2: Yes

Reviewer #3: Yes

Reviewer #4: Yes

5. Review Comments to the Author

Reviewer #1: Thank you for the opportunity to review this manuscript which presents the outcomes from an RCT comparing the efficacy of an mHealth intervention with the free Text4baby app to achieve weight loss in postpartum women with overweight or obesity and affected by a previous pregnancy complication. I have the following suggestions for authors below.

Introduction: the introduction is very well written, with clear background, rationale for study and objectives stated.

Methods: the methods are much less clearly written and overall would benefit from revision to improve clarity and logic. I have the following specific comments for consideration.

1. Ethical approval number – study approval is stated, but it is good practice to include the approval number, as well as the institute that gave approval

2. When/how was height measured? It says in statistical analysis that measured height was used to calculate BMI, but measurement of height is not specified in Measures section

3. Study participants should be described as ‘participants’ rather than ‘patients’, please check throughout manuscript

4. How was high risk pregnancy identified? Hospital notes? By physician?

5. Please give more detail about self-report diet and PA questionnaires. Are they validated questionnaires? What types of outcomes/data do they collect and record?

Results: revision of results is also recommended to improve clarity. Specific comments are as follows.

1. You state that Data not shown for Physical Activity results. However, could you please state overall activity levels of participants to give context to reader (even though there were no significant findings/ differences between groups).

2. For dietary intake data, you only show differences from baseline to 6/12 months. This gives no indication of what the intake actually was. This information is required to allow the reader to make any sort of interpretation of your findings. Could include baseline measure, if not in the table, in the text?

Discussion: logical flow and clarity of discussion could also be improved.

1. Can you place your findings within any literature concerning the amount of typical weight loss experienced during the first 6-12 months postpartum (i.e. without intervention), as women may experience gradual weight loss and return to pre-pregnancy weight postpartum without intervention.

2. In your discussion you refer to literature reporting that women have inadequate levels of physical activity postpartum. However, you have not stated that actual amount of physical activity by women in your study. Please add to findings to allow give this discussion point more context.

3. The conclusion for this study should answer to the aim of your study, i.e. that your study did not find a difference between the FAB and Text4baby groups. Also, please be careful not to overstate your findings, saying that engagement was ‘excellent’ may be considered an overstatement. Women only engaged with the weekly coaching sessions less than half of the time, and you were not able to compare intervention with control group engagement.

General comments:

1. There are several typos throughout the Methods, Results and Discussion, including line 254 and line 393

2. Person-first language should be used throughout (it is done in the introduction, but then discontinued throughout methods, results and discussion) to describe “women with overweight/obesity”, rather than “overweight/obese women”. See advice from Obesity Action Coalition: https://www.obesityaction.org/action-through-advocacy/weight-bias/people-first-language/

Reviewer #2: Dear Authors,

Thank you for the opportunity to review this interesting and valuable paper. I have some minor suggestions for changes below:

Introduction:

You assert the link between complications in pregnancy with adverse pregnancy/delivery/neonatal/postpartum outcomes, but it would be good to have some information on risk to subsequent pregnancies – for example, those with GDM are at an increased risk of GDM in any subsequent pregnancies. Recognition of the impact of these conditions and associated weight management issues on the interconception period would also be advantageous.

Methods:

Apologies if I missed this, but it would be good to understand if you excluded women who had a certain number of previous pregnancies. Also, did you exclude women pregnant with twins/triplets etc.?

It would be good to understand why in particular you have decided to approach women at 6 weeks postpartum. A line of justification would be good in this section.

There is no mention in this section of how you measure app acceptability/engagement or how it is analysed. Some addition of this information would substantially help with the claims made in the discussion around ‘excellent’ engagement.

Lines 168-169 – You mention an android coaching app, but before you say that FAB is only available on iPhone. This is confusing, were participants provided with another android phone? Related to this it would be good to understand who the coach was – a psychologist? Someone training in motivational interviewing?

Discussion

Lines 326-328 – I find this explanation confusing as the T4B intervention was not aimed at weight loss (making it an active control) – the breastfeeding explanation makes much more sense, perhaps the first sentence could be linked to the next point? E.g. the continuation in texts may have promoted behaviours such as breastfeeding which may have impacted weight.

Line 361 – how do you define ‘excellent’ engagement? How was this measured? Is this in comparison to the other studies you’ve detailed? It would be good to have some more detail here in regards to what your thoughts are on what it was about this app in particular that was so engaging? Your argument reads a bit like it was because it was an app, but there are plenty of apps out there that are not. For example, was it the gamification elements that make it stand out from other competitors?

Related to engagement it would be good to understand women’s compliance with things like charging the FitBit. Perhaps some qualitative work around the practicalities of this intervention is something you plan to include for future work?

Line 393 – I think the end of this sentence might be missing.

In your limitations section it would be good to see some discussion around retention. For example, 325 participants met eligibility but 154 consented. Do you know why? Out of the 154 consented only 82 attended the baseline visit. Some discussion about this drop would be useful. For example, is it possible that those who made it to the baseline visit were more motivated to manage their weight than those who did not attend? Is it possible that those who took part had greater resources to take part in research activities and therefore had more resources to manage their weight? I realise that you have some diversification in your population but the majority are college graduates and earn over $75,000.

Conclusions/summary – this section was lacking slightly. It would be good to understand your future directions and plans for future research here and any recommendations you might have for further app development. E.g. inclusion of gamification/wearables.

Reviewer #3: The authors report an mHealth intervention to support postpartum women who had experienced adverse pregnancy outcomes. This makes a significant contribution to research to improve the health and wellbeing of mothers and reduce risk factors for future cardiometabolic disease.

Comments:

Line 212 – It is preferable to write out dates clearly to avoid confusion about whether you are using MM/DD/YYYY or DD/MM/YYYY date convention. E.g, is recruitment between 9th April 2017 to 10th July 2019 or 4th September 2017 to 7th October 2019?

In the discussion, the authors did not discuss the possible long-term effects of this findings. For example, it seems that the benefits gained from the intervention was not sustained beyond the active phase of the intervention.

Can the authors discuss the implications of their finding of a sustained effect in the T4B group during pandemic which was absent from the FAB group? Does that suggest that low intervention dose over a long period may be equally as effective for this population as short intensive (12 weeks) intervention dose? Sustainability of weight loss and healthy lifestyle behaviours is important in this population. Although the authors attempted to explain why there were differences in intervention effects during the pandemic, they have not sufficiently discussed the possible implications of this (e.g., for the modification of the FAB intervention dose/intensity/duration). Perhaps there is not enough power to draw a conclusion here, but this should be acknowledged. Also, since this is an mHealth intervention, why would the pandemic have affected outcomes, since participants were not physically visiting any facility? This is not clear to the reader.

Line 393 is not complete.

Reviewer #4: The manuscript addresses an interesting topic. The collected data are unique and the employed statistical methods are generally sound, though more details are required. The results are consistent and offer a nice view also for further researches on the topic. Some comments follow.

1. The data are not fully available for the reviewers. This does not allow for the correctness of the methods and the replicability of the results. Moreover, it would be nice to have more details on the used statistical software/package/function to obtain the results; the code should be uploaded as supplementary material, along with the data (for review purposes only).

2. The statistical methods are generally sound. I really appreciate the use of mixed models. However, more details are required:

a) It is rather unclear how the linear predictor is specified. Are you considering a growth model? How do you account for the baseline effect as it is well known to affect the random effects distribution (see e.g. https://doi.org/10.1007/s11222-006-7072-5)? Please, write down the linear predictor to appreciate the model you fit to the data.

b) Missing mechanism may be completely at random, at random or not at random. It is rather unclear how missingnes is accounted for. Did you consider a pattern mixture or a selection model or...to deal with missing not at random?

c) The random effects distribution is often taken for granted and a Gaussian distribution is considered. I guess it is so also for you model. Please, provide evindence of the robustness of your results with respect to a misspecification of the random effects distribution.

d) All methods and parametric tests must fulfill some strict assumptions to avoid misleading inference. As data are not available, it is rather impossible to verify the adequacy of a linear model, rather than e.g. a heavy tails or a skew model. Please, provide evidence that all model's assumptions are met; the residual analysis would be helpful to clarify this point. This is also true for t-tests, whose main assumption is that the data follow a gaussian homoschedastic distribution; I guess you are considering paired t-tests, please clarify.

e) I am wondering if interactions may arise or if collinearity may be an issue. A discussion on variable selection would be helpful.

6. PLOS authors have the option to publish the peer review history of their article (what does this mean?). If published, this will include your full peer review and any attached files.

Reviewer #1: No

Reviewer #2: No

Reviewer #3: **Yes: **Dr Maureen Makama

Reviewer #4: No

---

## [Author Response · Author response to Decision Letter 0]

10 Aug 2023

Dear Editors,

We appreciate the reviews to our manuscript entitled, “The Fit After Baby randomized controlled trial: an mHealth postpartum lifestyle intervention for women with elevated cardiometabolic risk.” We have made every effort to address the issues raised.

We have addressed the additional requirements as follows:

We have edited the manuscript to follow PLOS ONE’s style requirements. 

2. Please expand the acronym “NHLBI” (as indicated in your financial disclosure) so that it states the name of your funders in full.

We have added National Heart Lung and Blood Institute to the cover letter so that it can be added to the online submission form. 

"We would like to thank the Fit After Baby participants. We would like to acknowledge the support from research assistants Danielle Cook, Chelsea Arent, Emily Dunn, and Jamie Siegart. We are very grateful for extensive contributions and support from Dean Hovey, Susan Gilbert, Sue Arment, and Glenn Bachmann. This study was supported by National Institutes of Health NIH (BIRCWH K12 HD057022 and NIH NHLBI 1K23HL133604) and NIH/National Center for Advancing Translational Sciences Colorado (CTSA UL1 TR002541). This trial is registered at ClinicalTrials.gov, NCT03215173 (https://clinicaltrials.gov/ct2/show/NCT03215173). "

"Author JMN was supported by three grants: 

- NIH BIRCWH K12 HD057022, National Institutes of Health, URL: https://orwh.od.nih.gov/career-development-education/building-interdisciplinary-research-careers-in-womens-health-bircwh

NIH NHLBI 1K23HL133604

https://www.nhlbi.nih.gov/grants-and-training/training-and-career-development/early-career

- NIH/NCATS Colorado CTSA UL1 TR002541, National Institutes of Health, National Center for Advancing Translational Sciences, URL: https://ncats.nih.gov/ctsa

We have taken funding information out of the acknowledgements section. There were three funding sources listed which match the three listed in the funding statement. We do not think we need to amend the statement. We have removed the clinicaltrials.gov number. 

We requested permission to share a de-indentified dataset from our governing body and we were told that we are not allowed to do so. We received the following response:

Because patient-level (e.g. “row-level” or “line-level”) data is more readily re-identifiable than summary data, CU Anschutz has a risk management policy of provisioning these data only via secure means such as NIH approved repositories (e.g. dbGap or other NIH clinical research registries: https://www.nih.gov/health-information/nih-clinical-research-trials-you/list-registries) or directly to investigators via secure campus data access control mechanisms. Summary level data and metadata about the patient-level data can be provided in support of the FAIR principles. 

Here are the contacts:

Melissa Haendel, PhD, FACMI

Chief Research Informatics Officer

MELISSA.HAENDEL@CUANSCHUTZ.EDU

Alison Lakin RN, LLB, LLM, PhD

Associate Vice Chancellor for Regulatory Compliance

ALISON.LAKIN@CUANSCHUTZ.EDU

We have added this to our cover letter. 

We have removed “data not shown,” and include the relevant tables in the manuscript. 

We have added the full name of our review board to our Methods section as follows:

The Colorado Multiple Institutional Review Board at the University of Colorado approved the study, and all patients gave written informed consent. (lines 155-157)

7. We note that the original protocol file you uploaded contains a confidentiality notice indicating that the protocol may not be shared publicly or be published. Please note, however, that the PLOS Editorial Policy requires that the original protocol be published alongside your manuscript in the event of acceptance. Please note that should your paper be accepted, all content including the protocol will be published under the Creative Commons Attribution (CC BY) 4.0 license, which means that it will be freely available online, and any third party is permitted to access, download, copy, distribute, and use these materials in any way, even commercially, with proper attribution.

Therefore, we ask that you please seek permission from the study sponsor or body imposing the restriction on sharing this document to publish this protocol under CC BY 4.0 if your work is accepted. We kindly ask that you upload a formal statement signed by an institutional representative clarifying whether you will be able to comply with this policy. Additionally, please upload a clean copy of the protocol with the confidentiality notice (and any copyrighted institutional logos or signatures) removed.

The Colorado Multiple Institutional Review Board at the University of Colorado approved an amendment for a publishable protocol. We include a letter from the Colorado Multiple Institutional Review Board at the University of Colorado stating that we may share this protocol publicly. We have included a publishable protocol with our revision.

Review Comments to the Author

Reviewer #1: Thank you for the opportunity to review this manuscript which presents the outcomes from an RCT comparing the efficacy of an mHealth intervention with the free Text4baby app to achieve weight loss in postpartum women with overweight or obesity and affected by a previous pregnancy complication. I have the following suggestions for authors below.

Introduction: the introduction is very well written, with clear background, rationale for study and objectives stated.

Methods: the methods are much less clearly written and overall would benefit from revision to improve clarity and logic. I have the following specific comments for consideration.

We have reordered and added headings to the Methods to improve clarity and logic. 

1. Ethical approval number – study approval is stated, but it is good practice to include the approval number, as well as the institute that gave approval

We have added the full name of our review board to our Methods section as follows:

The Colorado Multiple Institutional Review Board at the University of Colorado approved the study (17-0045), and all patients gave written informed consent. (lines 155-157)

2. When/how was height measured? It says in statistical analysis that measured height was used to calculate BMI, but measurement of height is not specified in Measures section

We have added information about the measurement of weight and height to the manuscript in the Methods section. 

At each visit trained staff measured body weight twice wearing light clothing, and weights were averaged (SECA 360), and height was measured by stadiometer (SECA). We used kg/m2 to determine BMI. Trained staff also measured waist circumference. (lines 181-183)

3. Study participants should be described as ‘participants’ rather than ‘patients’, please check throughout manuscript

We have changed ‘patients’ to ‘participants’ throughout the manuscript.

4. How was high risk pregnancy identified? Hospital notes? By physician?

We have added the following to the Methods section:

We identified patients by diagnosis codes, and pregnancy complications were confirmed via chart review by the study physician. (lines 144-145)

5. Please give more detail about self-report diet and PA questionnaires. Are they validated questionnaires? What types of outcomes/data do they collect and record?

The validated diet questionnaire we used was the 2005 Block Food Frequency Questionnaire (administered via NutritionQuest). This questionnaire provides an estimate of habitual intake. The validated physical activity questionnaire was the Pregnancy Physical Activity Questionnaire (PPAQ). This questionnaire provides a reasonably accurate measure of a broad range of physical activities. 

Results: revision of results is also recommended to improve clarity. Specific comments are as follows.

1. You state that Data not shown for Physical Activity results. However, could you please state overall activity levels of participants to give context to reader (even though there were no significant findings/ differences between groups).

We have added the data for the physical activity results and summarized them in the Results section. 

2. For dietary intake data, you only show differences from baseline to 6/12 months. This gives no indication of what the intake actually was. This information is required to allow the reader to make any sort of interpretation of your findings. Could include baseline measure, if not in the table, in the text?

We have added a table of baseline measures to the manuscript. 

Discussion: logical flow and clarity of discussion could also be improved.

1. Can you place your findings within any literature concerning the amount of typical weight loss experienced during the first 6-12 months postpartum (i.e. without intervention), as women may experience gradual weight loss and return to pre-pregnancy weight postpartum without intervention.

-We have included the following on typical weight loss in postpartum women in the Discussion.

Although there is a lot of variability in postpartum weight loss, studies show that 15-27% of women have major postpartum weight retention at one year of at least 4.55 kg. Nearly all women (97%) who have obesity before pregnancy will continue to be classified as such at one year, with 40% increasing by two or more BMI units. Among women with overweight, 40-50% will move into the obesity category by 12 months postpartum. (lines 474-479)

2. In your discussion you refer to literature reporting that women have inadequate levels of physical activity postpartum. However, you have not stated that actual amount of physical activity by women in your study. Please add to findings to allow give this discussion point more context.

-We have included baseline data on physical activity in our population.

3. The conclusion for this study should answer to the aim of your study, i.e. that your study did not find a difference between the FAB and Text4baby groups. Also, please be careful not to overstate your findings, saying that engagement was ‘excellent’ may be considered an overstatement. Women only engaged with the weekly coaching sessions less than half of the time, and you were not able to compare intervention with control group engagement.

We have changed the conclusion to address our primary outcome and the failure to reach significance. We have changed our assessment of engagement to be “promising.” We have also edited the abstract to be consistent with these conclusions. 

General comments:

1. There are several typos throughout the Methods, Results and Discussion, including line 254 and line 393

We have corrected the typos on line 254 and removed the sentence in 393. 

2. Person-first language should be used throughout (it is done in the introduction, but then discontinued throughout methods, results and discussion) to describe “women with overweight/obesity”, rather than “overweight/obese women”. See advice from Obesity Action Coalition: https://www.obesityaction.org/action-through-advocacy/weight-bias/people-first-language/

We have revised the manuscript to use person-first language throughout.

Reviewer #2: Dear Authors,

Thank you for the opportunity to review this interesting and valuable paper. I have some minor suggestions for changes below:

Introduction:

You assert the link between complications in pregnancy with adverse pregnancy/delivery/neonatal/postpartum outcomes, but it would be good to have some information on risk to subsequent pregnancies – for example, those with GDM are at an increased risk of GDM in any subsequent pregnancies. Recognition of the impact of these conditions and associated weight management issues on the interconception period would also be advantageous.

-In the introduction we have added data from the literature on the importance of weight loss for the interconception period on GDM and other pregnancy complications in subsequent pregnancies. (lines 91-94)

Methods:

Apologies if I missed this, but it would be good to understand if you excluded women who had a certain number of previous pregnancies. Also, did you exclude women pregnant with twins/triplets etc.?

-We excluded all non-singleton pregnancies for this study. There were no exclusion criteria based upon the number of previous pregnancies. We have added this to the methods section. 

It would be good to understand why in particular you have decided to approach women at 6 weeks postpartum. A line of justification would be good in this section.

-We decided to start the trial at 6 weeks postpartum because this is typically when women will return for their postpartum visit, and women with a pregnancy complicated by gestational diabetes should have postpartum glucose testing at this time. Our previous qualitative work and previous studies demonstrated that this is a reasonable time to begin a lifestyle intervention in postpartum women. We have added this information and citations to the manuscript.

There is no mention in this section of how you measure app acceptability/engagement or how it is analysed. Some addition of this information would substantially help with the claims made in the discussion around ‘excellent’ engagement.

-We have added information about how we measured app engagement to the Methods section. We collected data on use of the app, including the number of days the app was opened, which content was opened, collected data on app use, steps and minutes of physical activity, days activity trackers were worn, and number of coaching interactions, and Health Warrior points accumulated. Usage data were collected in BigQuery (Google). App usage data were analyzed using BigQuery (Google) and Tableau (Mountain View, CA).

Lines 168-169 – You mention an android coaching app, but before you say that FAB is only available on iPhone. This is confusing, were participants provided with another android phone? Related to this it would be good to understand who the coach was – a psychologist? Someone training in motivational interviewing?

-The coaching app was built on an Android platform. This was purely based on the coding preference of the coders who built the app. Consequently the coach used an Android platform to access the coaching data. Participants used iPhones for access to the app. Since the reference to the Android platform is confusing and not essential to the discussion we have removed this. The coach was a registered dietitian with training in motivational interviewing. We have added this to the coaching section of the methods.

Discussion

Lines 326-328 – I find this explanation confusing as the T4B intervention was not aimed at weight loss (making it an active control) – the breastfeeding explanation makes much more sense, perhaps the first sentence could be linked to the next point? E.g. the continuation in texts may have promoted behaviours such as breastfeeding which may have impacted weight.

-This is a good point and I have linked these concepts in the text. In addition, the ongoing texts from T4B, particularly in the early part of the pandemic, may have led to an increased sense of connection for those in the control group. 

Line 361 – how do you define ‘excellent’ engagement? How was this measured? Is this in comparison to the other studies you’ve detailed? It would be good to have some more detail here in regards to what your thoughts are on what it was about this app in particular that was so engaging? Your argument reads a bit like it was because it was an app, but there are plenty of apps out there that are not. For example, was it the gamification elements that make it stand out from other competitors?

-We have re-written this to describe the engagement as “promising.” We believe that the engagement is better than many similar technologically-based interventions in this population, but agree that it is not to the level of excellent. We have added more detail about the way the user data were collected and analyzed. We have added more data on the attainment of reward badges to further explain the gamification component. 

Related to engagement it would be good to understand women’s compliance with things like charging the FitBit. Perhaps some qualitative work around the practicalities of this intervention is something you plan to include for future work?

-We do not have quantitative data on Fitbit charging in this study. We did do focus groups at the conclusion of this study and we are in the process of analyzing these data. 

Line 393 – I think the end of this sentence might be missing.

-we have removed this sentence. 

In your limitations section it would be good to see some discussion around retention. For example, 325 participants met eligibility but 154 consented. Do you know why? Out of the 154 consented only 82 attended the baseline visit. Some discussion about this drop would be useful. For example, is it possible that those who made it to the baseline visit were more motivated to manage their weight than those who did not attend? Is it possible that those who took part had greater resources to take part in research activities and therefore had more resources to manage their weight? I realise that you have some diversification in your population but the majority are college graduates and earn over $75,000.

-We have added some discussion around retention to the limitations section. We do not have detailed data on why those who met eligibility were not consented. Among those who consented we know that among the 57 women we either could not schedule them or they were no longer interested. We agree with the importance of discussing the ways those who decided to participate may be different from those who did not and we have added this to the manuscript. 

Conclusions/summary – this section was lacking slightly. It would be good to understand your future directions and plans for future research here and any recommendations you might have for further app development. E.g. inclusion of gamification/wearables.

-We have edited the conclusion to better represent the results and to discuss plans for improvements that could enhance efficacy. 

Reviewer #3: The authors report an mHealth intervention to support postpartum women who had experienced adverse pregnancy outcomes. This makes a significant contribution to research to improve the health and wellbeing of mothers and reduce risk factors for future cardiometabolic disease.

Comments:

Line 212 – It is preferable to write out dates clearly to avoid confusion about whether you are using MM/DD/YYYY or DD/MM/YYYY date convention. E.g, is recruitment between 9th April 2017 to 10th July 2019 or 4th September 2017 to 7th October 2019?

We have written out dates throughout the manuscript. 

In the discussion, the authors did not discuss the possible long-term effects of this findings. For example, it seems that the benefits gained from the intervention was not sustained beyond the active phase of the intervention.

Can the authors discuss the implications of their finding of a sustained effect in the T4B group during pandemic which was absent from the FAB group? Does that suggest that low intervention dose over a long period may be equally as effective for this population as short intensive (12 weeks) intervention dose? 

-This is an important point. The number of Text4baby participants during the pandemic was only 6, so it is difficult to draw a lot of conclusions from their data. It is possible that women in the Text4baby group, who were still receiving 3-4 texts per week, may have had a greater sense of connection. Text4baby may have promoted behaviors such as continuation of breastfeeding leading to increased weight loss. In addition, women in the control group were significantly more likely to be breastfeeding at baseline, which may have influenced their weight loss. We have added more to the discussion on this topic. A larger study would be needed to test the difference between a low intervention dose over a long period vs. a more intensive intervention over a shorter period. We have added more to the manuscript to reflect this. 

Sustainability of weight loss and healthy lifestyle behaviours is important in this population. Although the authors attempted to explain why there were differences in intervention effects during the pandemic, they have not sufficiently discussed the possible implications of this (e.g., for the modification of the FAB intervention dose/intensity/duration). Perhaps there is not enough power to draw a conclusion here, but this should be acknowledged. Also, since this is an mHealth intervention, why would the pandemic have affected outcomes, since participants were not physically visiting any facility? This is not clear to the reader.

-We have added further ideas about the implications of dose and intensity and duration. We believe that the pandemic affected lifestyles for the women in our study in that it impacted diet and exercise and weight. We believe that the impact of the pandemic overwhelmed what lifestyle changes could have been driven by the Fit After Baby program, particularly after the active part of the intervention had finished. Even though the intervention was remote, the more intensive/interactive part of the program was finished for the FAB participants before the start of the COVID pandemic. 

Line 393 is not complete.

-We have removed line 393.

Reviewer #4: The manuscript addresses an interesting topic. The collected data are unique and the employed statistical methods are generally sound, though more details are required. The results are consistent and offer a nice view also for further researches on the topic. Some comments follow.

1. The data are not fully available for the reviewers. This does not allow for the correctness of the methods and the replicability of the results. Moreover, it would be nice to have more details on the used statistical software/package/function to obtain the results; the code should be uploaded as supplementary material, along with the data (for review purposes only).

-We were not given permission to upload the dataset at the individual level. We used SAS software (SAS Inc., Cary, NC), version 9.4 for analysis, as stated in the statistical methods of the manuscript. 

2. The statistical methods are generally sound. I really appreciate the use of mixed models. 

Thank you for the positive review of our work.

However, more details are required:

a) It is rather unclear how the linear predictor is specified. Are you considering a growth model? How do you account for the baseline effect as it is well known to affect the random effects distribution (see e.g. https://doi.org/10.1007/s11222-006-7072-5)? Please, write down the linear predictor to appreciate the model you fit to the data.

Thank you for your comment. We used a mixed-effects model with a random intercept. The general form of the mixed-effects models that we used is as follows:

Y_ij= β_0+ β_1 X_ij+ u_j+ e_j

where u_j ~ N(0,σ_u^2 ) is the random intercept. Our models were adjusted for the baseline value of the outcome. We also noted that the reference you provided referred to models for longitudinal count data and therefore would not apply to our data.

b) Missing mechanism may be completely at random, at random or not at random. It is rather unclear how missingnes is accounted for. Did you consider a pattern mixture or a selection model or...to deal with missing not at random?

We agree with the reviewer that consideration of missing data is an important part of all statistical analyses. By using mixed-effects models, we were able to include all outcome data regardless of whether a participant had one or more missing values. The amount of missing data in our study was relatively low and therefore was unlikely to cause significant bias even if the assumption of missing completely at random/missing at random was violated, so we did not consider pattern mixture models or other models to account for missing data. We also performed a sensitivity analysis by comparing the results of our analyses using all available data with one using data only from participants with complete data, and our findings were unchanged; therefore, we concluded that the amount of missing data in this study was not likely to introduce errors in our conclusions.

c) The random effects distribution is often taken for granted and a Gaussian distribution is considered. I guess it is so also for you model. Please, provide evidence of the robustness of your results with respect to a misspecification of the random effects distribution.

To evaluate the assumption of the normality of the random effects, we created histograms of the model residuals using the BLUP (best unbiased linear prediction) estimates for the models of our primary outcome, weight. These plots (provided below) do not show any evidence of significant non-normality. Note that the plot labeled “analweight_kg” comes from a model using the post-partum week 6 visit as the baseline, and the plot labeled “analweight2_kg” comes from a model using pre-pregnancy weight as the baseline.

d) All methods and parametric tests must fulfill some strict assumptions to avoid misleading inference. As data are not available, it is rather impossible to verify the adequacy of a linear model, rather than e.g. a heavy tails or a skew model. Please, provide evidence that all model's assumptions are met; the residual analysis would be helpful to clarify this point. This is also true for t-tests, whose main assumption is that the data follow a gaussian homoschedastic distribution; I guess you are considering paired t-tests, please clarify.

Please see our response above regarding the residuals from the linear mixed-effects models. Our primary analyses used mixed-effects models rather than t-tests. The only t-tests reported in the paper are two-sample t-tests in Table 1. Because we had a sample size of 81 women, the results of the t-tests in Table 1 are most likely robust to deviations from assumptions.

See: Posten, H.O., Yeh, H.C. and Owen, D.B. (1982): Robustness of the two-sample t-test under violations of the homogeneity of variance assumptions. Communications in Statistics: Theory and Methods 11, 109-126

e) I am wondering if interactions may arise or if collinearity may be an issue. A discussion on variable selection would be helpful. 

Thank you for your question. Variables in our models were selected a priori based on subject matter expertise. We did not hypothesize that there would be important interactions, and in order to minimize the amount of statistical testing and therefore the risk of type I error, we did not test interactions of pairwise combinations of covariates. There was no evidence of collinearity (e.g., inflated standard errors for parameter estimates) in our model results.

---

## [Decision Letter · Decision Letter 1]

4 Sep 2023

PONE-D-23-03981R1The Fit After Baby randomized controlled trial: An mHealth postpartum lifestyle intervention for women with elevated cardiometabolic riskPLOS ONE

Dear Dr. Nicklas,

Thank you for submitting your manuscript to PLOS ONE. After careful consideration, we feel that it has merit but does not fully meet PLOS ONE’s publication criteria as it currently stands. Therefore, we invite you to submit a revised version of the manuscript that addresses the points raised during the review process.

 In particular, please ensure that the reviewer's comments pertaining to the statistical modelling used in your analyses are addressed, including justification of results obtained using the analyses conducted. 

We look forward to receiving your revised manuscript.

Kind regards,

Megan L Gow

Guest Editor

PLOS ONE

Reviewers' comments:

Reviewer's Responses to Questions

**Comments to the Author**

1. If the authors have adequately addressed your comments raised in a previous round of review and you feel that this manuscript is now acceptable for publication, you may indicate that here to bypass the “Comments to the Author” section, enter your conflict of interest statement in the “Confidential to Editor” section, and submit your "Accept" recommendation.

Reviewer #3: All comments have been addressed

Reviewer #4: (No Response)

2. Is the manuscript technically sound, and do the data support the conclusions?

Reviewer #3: Yes

Reviewer #4: Partly

3. Has the statistical analysis been performed appropriately and rigorously? 

Reviewer #3: Yes

Reviewer #4: No

4. Have the authors made all data underlying the findings in their manuscript fully available?

Reviewer #3: Yes

Reviewer #4: No

5. Is the manuscript presented in an intelligible fashion and written in standard English?

Reviewer #3: Yes

Reviewer #4: Yes

6. Review Comments to the Author

Reviewer #3: (No Response)

Reviewer #4: Thank you very much for all the efforts to reply to my previous comments. Some clarifications are still required.

1. You state that "We also noted that the reference you provided referred to models for longitudinal count data and therefore would not apply to our data." This is far from being true. Your model casts into the generalized linear mixed framework, as well as the provided reference. It is well-known that the initial conditions may strongly bias the estimated coefficients, see also the Heckman model. Accordingly, the model must be modified accordingly and results compared with those presented in the current version of the paper.

2. According to your reply about the missing data mechanism is at random. This assumption is hardly tenable. Moreover, the nature of missing data mechanism does not depend on the amount of missingnes, please refer to the statistical literature on the topic and to Little's work in particular.

3. Please, provide more details on the check of model's assumptions. It is rather unclear why the BLUP graph should provide information about the random effects distribution. Similarly, having a sample of 80 or more observations does not guarantee that normality is met. Even one outlying observation only may strongly affect the results.

7. PLOS authors have the option to publish the peer review history of their article (what does this mean?). If published, this will include your full peer review and any attached files.

Reviewer #3: No

Reviewer #4: No

---

## [Author Response · Author response to Decision Letter 1]

15 Nov 2023

Dear reviewers,

Thank you for your help improving the manuscript thus far. Below please find our responses to the latest round of comments. 

Reviewer #4: Thank you very much for all the efforts to reply to my previous comments. Some clarifications are still required.

1. You state that "We also noted that the reference you provided referred to models for longitudinal count data and therefore would not apply to our data." This is far from being true. Your model casts into the generalized linear mixed framework, as well as the provided reference. It is well-known that the initial conditions may strongly bias the estimated coefficients, see also the Heckman model. Accordingly, the model must be modified accordingly and results compared with those presented in the current version of the paper.

Thank you for your comment; that is an excellent point. We have re-run all of our models including baseline values as a covariate. This led to some changes in our results and we have rewritten our manuscript accordingly. We have also rewritten the Methods to state that we have run our models with baseline as a covariates, and we have added this as a footnote to all tables. 

2. According to your reply about the missing data mechanism is at random. This assumption is hardly tenable. Moreover, the nature of missing data mechanism does not depend on the amount of missingnes, please refer to the statistical literature on the topic and to Little's work in particular.

Thank you for the opportunity to clarify. We did not mean to imply that the nature of the missing data mechanism depended on the amount of missing data, but rather that because the amount of missing data was relatively low, it would be less likely to impact our conclusions even if the assumption of missing completely at random or missing at random was violated.

The table below provides a comparison of the baseline characteristics and randomization assignments of participants who did and did not have missing data. These two groups are quite similar, further supporting the idea that missing data in this study was unlikely to cause significant bias.

Baseline Characteristics Missing data 

 Yes No P values

Age 29.4 (5.4) 31.3 (5.4) 0.1639

BMI 32.9 (4.9) 32.1 (5.0) 0.5241

Weight(Kg) 89.0 (15.4) 86.8 (14.8) 0.5369

Group 0.8795

 Intervention 16 (66.7%) 37 (64.9%) 

 Control 8 (33.3%) 20 (35.1%) 

Affected by COVID 0.4974

 Yes 5 (20.8%) 16 (28.1%) 

 No 19 (79.2%) 41 (71.9%) 

Finally, because this is an RCT, conditioning on the baseline value of the outcome reduces the potential for bias (PMID: 22262640). We have added a sentence about this comparison to the Results section and a sentence to the limitations. 

3. Please, provide more details on the check of model's assumptions. It is rather unclear why the BLUP graph should provide information about the random effects distribution. Similarly, having a sample of 80 or more observations does not guarantee that normality is met. Even one outlying observation only may strongly affect the results.

Thank you for the opportunity to clarify. The BLUPs are an estimate of the random effects. Please see the following reference:

Christian Ritz. (2004). Goodness-of-Fit Tests for Mixed Models. Scandinavian Journal of Statistics, 31(3), 443–458. http://www.jstor.org/stable/4616841

---

## [Decision Letter · Decision Letter 2]

10 Dec 2023

The Fit After Baby randomized controlled trial: An mHealth postpartum lifestyle intervention for women with elevated cardiometabolic risk

PONE-D-23-03981R2

Dear Dr. Nicklas,

We’re pleased to inform you that your manuscript has been judged scientifically suitable for publication and will be formally accepted for publication once it meets all outstanding technical requirements.

Kind regards,

Megan L Gow

Guest Editor

PLOS ONE

Additional Editor Comments (optional):

Reviewers' comments:

Reviewer's Responses to Questions

**Comments to the Author**

1. If the authors have adequately addressed your comments raised in a previous round of review and you feel that this manuscript is now acceptable for publication, you may indicate that here to bypass the “Comments to the Author” section, enter your conflict of interest statement in the “Confidential to Editor” section, and submit your "Accept" recommendation.

Reviewer #4: All comments have been addressed

2. Is the manuscript technically sound, and do the data support the conclusions?

Reviewer #4: (No Response)

3. Has the statistical analysis been performed appropriately and rigorously? 

Reviewer #4: (No Response)

4. Have the authors made all data underlying the findings in their manuscript fully available?

Reviewer #4: (No Response)

5. Is the manuscript presented in an intelligible fashion and written in standard English?

Reviewer #4: (No Response)

6. Review Comments to the Author

Reviewer #4: (No Response)

7. PLOS authors have the option to publish the peer review history of their article (what does this mean?). If published, this will include your full peer review and any attached files.

Reviewer #4: No

---

## [Editor Report · Acceptance letter]

27 Dec 2023

PONE-D-23-03981R2 

PLOS ONE

Dear Dr. Nicklas, 

I'm pleased to inform you that your manuscript has been deemed suitable for publication in PLOS ONE. Congratulations! Your manuscript is now being handed over to our production team.

Kind regards, 

on behalf of

Dr. Megan L Gow 

Guest Editor

PLOS ONE